



# snowScatt 1.0: Consistent model of microphysical and scattering properties of rimed and unrimed snowflakes based on the self-similar Rayleigh-Gans Approximation

Davide Ori[a], Leonie von Terzi[a], Markus Karrer[a], and Stefan Kneifel[a]

[a]Institute for Geophysics and Meteorology, University of Cologne, Cologne, Germany

**Correspondence:** Davide Ori (dori@uni-koeln.de)

**Abstract.** More detailed observational capabilities in the microwave (MW) and advancements in the details of microphysical schemes for ice and snow demand increasing complexity to be included in scattering databases. The majority of existing databases rely on the Discrete Dipole Approximation (DDA) whose high computational costs limit either the variety of particle types or the range of parameters included, such as frequency, temperature, or particle size.

snowScatt is an innovative tool that provides the consistent microphysical and scattering properties of an ensemble of 50 thousand snowflake aggregates generated with different physical particle models. Many diverse snowflake types, including rimed particles and aggregates of different monomer composition, are accounted for. The scattering formulation adopted by snowScatt is based on the Self-Similar Rayleigh-Gans Approximation (SSRGA) which is capable of modeling the scattering properties of large ensembles of particles. Previous comparisons of SSRGA and DDA are extended in this study by including

unrimed and rimed aggregates up to cm-sizes and frequencies up to the sub-mm spectrum. The results reveal in general the wide applicability of the SSRGA method for active and passive MW applications. Unlike DDA databases, the set of SSRGA coefficients can be used to infer the scattering properties at any frequency and refractive index. snowScatt also provides tools to derive the SSRGA coefficients for new sets of particle structures which can be easily included in the library.

The flexibility of the snowScatt tool with respect to applications that require continuously changing definitions of snow

properties is demonstrated in a forward simulation example based on the output of the Predicted Particle Properties (P3) scheme. snowScatt provides the same level of flexibility as commonly used T-matrix solutions while the computed scattering properties reach the level of accuracy of detailed Discrete Dipole Approximation calculations.

## 1   Introduction

Accurate characterization of scattering and absorption properties of hydrometeors in the microwave (MW) is an essential

prerequisite for retrievals of cloud and precipitation properties (Maahn et al., 2020). While the scattering properties for liquid hydrometeors are relatively well known, large uncertainties are still associated with frozen hydrometeors (Kneifel et al., 2020). Those uncertainties are currently also one of the main obstacles for assimilating space-borne MW observations under all-sky conditions (Kulie et al., 2010; Geer and Baordo, 2014; Geer et al., 2018).





As pointed out by Kneifel et al. (2020) and Tyynelä and von Lerber (2019), the problem of realistically characterizing the
scattering properties of ice crystals, snowflakes, and rimed particles is two-fold: First, the physical properties, such as size,
mass, density, shape, internal structure, or composition of ice and liquid has to be characterized. This can be done either
empirically or by using a physical hydrometeor model, which generates the particles by directly simulating a certain growth
process such as aggregation. A common model for snowflakes and rimed aggregates composed of various monomer types
was provided by (Leinonen and Szyrmer, 2015); the model has recently been extended, to also provide mixtures of various
monomer types for the generation of aggregates (Karrer et al., 2020). Second, once the particle properties are well defined, the
scattering properties can be derived with various numerical solvers. One of the most common methods is the Discrete Dipole
Approximation (DDA Draine and Flatau, 1994), where the particle structure is discretized on a regular, three-dimensional grid.
The DDA takes interactions of the scattering elements with the incident wave but also among each other into account. The
high accuracy of the DDA method (Yurkin et al., 2006; Ori and Kneifel, 2018) comes at the cost of the high complexity of the
calculations that have to be performed separately for each particle type, size, orientation, frequency, and temperature. Also the
resolution of the discretization and hence the number of scattering elements has to be enhanced for larger size parameters and
refractive indices of the particle in order to keep the uncertainties in the scattering properties low.

During recent years, the number of scattering databases and the complexity of included particles has strongly increased
(Kneifel et al., 2018; Tyynelä and von Lerber, 2019). While earlier databases only included idealized ice crystals with random
orientation (e.g. Liu, 2008), more recent databases provide scattering properties for various particle orientations (Lu et al.,
2016), and also extensive frequency ranges up to the sub-mm region (Brath et al., 2020). A comprehensive comparison of 9
recent databases with in-situ, multi-frequency, and polarimetric radar observations collected in Finland is provided in Tyynelä
and von Lerber (2019). Overall, the best match for physical particle properties and scattering signatures were found for the
physical snow models.
The signals in active and passive MW observations are generally related to higher moments of the particle size distribution
(e.g., radar reflectivity factor $\propto$ 6th moment). Therefore, scattering properties of larger particles, such as aggregates or rimed
aggregates, have a strong impact on the signal even though their concentration might be relatively small. Recent studies revealed
that the physical (e.g., terminal velocity - size relation) and scattering properties (e.g., triple-frequency radar signatures) of
aggregates can depend on the monomer type and monomer size distribution (Leinonen and Moisseev, 2015; Karrer et al.,
2020). However, due to the high computational costs of deriving scattering properties for aggregates of large sizes with DDA,
most databases incorporate only few aggregate types.

Considering that any remote sensor is always measuring bulk scattering properties of an ensemble of particles, a better char-
acterization of ensemble scattering properties is desirable. The self-similar Rayleigh Gans Approximation (SSRGA Hogan and
Westbrook, 2014; Hogan et al., 2017) represents a new approach, which takes this aspect into account. The SSRGA method
is based on the classical Rayleigh-Gans approximation (Bohren and Huffman, 1983) where the interactions of the scattering
elements among each other are neglected and the resulting scattered electromagnetic field is calculated by integrating the con-
tributions of each scattering element independently. The SSRGA extends the RGA by decomposing the mass distributions of
an ensemble of self-similar snow aggregates with their statistical mean and the power spectrum of the fluctuations around this





mean distribution (see also Section 2.2). After a set of coefficients has been derived from the particle ensemble, the scatter-
ing parameters, such as phase function or cross-sections, can be calculated with analytical formulas for any frequency, size, or
temperature range. The main limitation of the SSRGA is the implicit assumption of a Rayleigh distribution of polarimetric com-
ponents. It is also expected that for particles with higher density, the interactions of scattering elements become non-negligible
(Westbrook et al., 2006). However, comparisons of DDA and SSRGA calculations for ensembles of rimed aggregates revealed
that a substantial bias can only be found at very high degrees of riming (Leinonen et al., 2018b). For forward simulations of
MW observations, for example using the output of numerical weather prediction models, it is often necessary to achieve a
consistency in the ice particle properties assumed in the model microphysics and in the forward model. Those properties, such
as mass-size relations, are often fixed in DDA based databases. The consistency problem becomes even larger for modelling
approaches, where the hydrometeor properties can change continuously, such as the Predicted Particle Properties scheme (P3
Morrison and Milbrandt, 2015), the morphology-predicting scheme of Tsai and Chen (2020) or Lagrangian super-particle
models (Brdar and Seifert, 2018; Shima et al., 2020). In order to achieve consistency between model and forward operator,
spheroidal models such as the Mie theory (Mie, 1908) or T-Matrix (Waterman, 1965) are still frequently used as the particle
properties can be adjusted here with varying the effective density of the spheroids. However, the underlying effective medium
approximation for calculating the refractive index of the homogeneous ice-air mixture has been found in several studies to
introduce inconsistent scattering properties, especially when a larger frequency range is considered (e.g. Geer and Baordo,
75   2014).

The SSRGA does not require an effective medium approximation as the distribution of mass within the particle is explicitly
parametrized. Principally, the mass-size relation is not fixed for SSRGA and it can be varied for a set of SSRGA coefficients.
The SSRGA coefficients can be derived with relatively low computational costs for a large variety of aggregate structures (e.g.
Mason et al., 2019). The effort to derive new coefficients for different particle orientations or ice refractive indices is much
lower than compared to DDA, where the complete simulations have to be repeated for every particle shape. Various scattering
properties obtained with SSRGA have been compared with DDA simulations (Hogan et al., 2017; Leinonen et al., 2018b;
Mech et al., 2020) and revealed a very good agreement even for moderately rimed particles. The scattering properties derived
with SSRGA also showed a very good agreement with the observed multi-frequency radar signatures of snowfall (Mason et al.,
2019; Dias Neto et al., 2019; Ori et al., 2020a).

Previously published SSRGA coefficients have been derived for slightly different formulations of the SSRGA and not all
of them provide the physical particle properties of the ensemble. In this study, we present a new software tool snowScatt,
which aims to simplify the application of the SSRGA method for the scientific community. It provides a database of previously
derived SSRGA coefficients as well as new coefficients based on a large aggregate database generated at University of Cologne
(Karrer et al., 2020). The current version of snowScatt includes the SSRGA coefficients derived for approximately 50 thousand
aggregates including various monomer types, unrimed, and rimed aggregates. In addition to the scattering properties, the tool
provides the associated microphysical properties, such as size, mass, area, and derived terminal fall velocity. snowScatt also
gives the possibility to derive SSRGA coefficients from an individual ensemble of three-dimensional particle structures, which





can then be added to snowScatt's coefficient library. Finally, snowScatt also includes a simple simulator for radar Doppler spectra and moments based on user-defined PSD.

In Sect. 2, we will shortly introduce into the theoretical foundations of the Self-Similar Rayleigh-Gans Approximation. Section 3 will provide an overview of the snowScatt package, including a description of the various aggregate types included and a comparison of their physical and scattering properties. Although the aim of this study is not a thorough evaluation of SSRGA, in Sect. 4 we will discuss the upper frequency and size limits up to which the SSRGA method can be reliably applied dependent on the aggregate type used. Also, we will show the advantage of using SSRGA ensemble properties with

respect to limited DDA databases. An application example of the snowScatt tool is provided in Sect. 5, where synthetic radar observations are simulated using spheroids, SSRGA, and one specific DDA particle habit based on model output generated with the P3 microphysical scheme. A short summary and outlook for future developments and applications of snowScatt are provided in Sect. 6.

## 2   Theoretical Background

### 2.1   The Rayleigh-Gans Approximation for single particles

The basis of the SSRGA methodology is the Rayleigh-Gans Approximation (RGA) which applies to "optically soft" particles. This condition states that the various parts of an arbitrarily-shaped particle only interact with the incident wave and the coupling among its scattering elements can be neglected. As a result, the scattered wave is the simple superposition of the individual contribution of each scattering element that behaves as a simple Rayleigh scatterer (Bohren and Huffman, 1983).

The conditions for the applicability of RGA are met when the refractive index of the scattering particle is not too different from the one of air. Also the size of the scatterer along the propagation direction of the incident wave should not be much larger than the wavelength. These two conditions are expressed mathematically as:

$$|n-1| \ll 1 \tag{1}$$

$$2kD|n-1| \ll 1 \tag{2}$$

where $n$ is the complex index of refraction, $k = 2\pi/\lambda$ is the wavenumber ($\lambda$ is the electromagnetic wavelength) and $D$ is the size of the scattering particle. For snowflake aggregates it is generally assumed that the combination of a relatively low

refractive index ($|n| \approx 1.78$ in the MW) and a very porous internal structure leads to the validity of the RGA assumptions (Sorensen, 2001). The second criterion (Eq. 2) explicitly depends on the scattering size parameter $x = kD$ ($D$ is the size of the particle along the propagation direction of the incident wave)

    Using RGA, the intensity of the scattered radiation $\sigma$ at an angle $\theta$ is given by the following formula (Hogan et al., 2017)

$$\sigma(\theta) = \frac{9}{4\pi} k^4 |K|^2 V^2 \frac{1 + \cos^2(\theta)}{2} \phi(x \sin(\theta/2)) \tag{3}$$

where $|K|^2$ is the Rayleigh dielectric factor, $V$ is the particle volume and $\phi$ is the so called form factor. The form factor is a

dimensionless value that accounts for the deviations of the scattering intensity from the pure Rayleigh approximation. Under



the RGA approximation the form factor is the integral over the particle volume of the phase delays among all the parts of the particle.

$$\phi_{\mathrm{RGA}}(x) = \frac{1}{V} \int\limits_{V} \exp(i\mathbf{R}(\mathbf{k}_{inc} - \mathbf{k}_{sca}))\mathrm{d}\mathbf{R} \tag{4}$$

Here, $\mathbf{k}_{inc}$ and $\mathbf{k}_{sca}$ denotes, respectively, the incident and scattering wavevector, while $\mathbf{R}$ is the position vector (from and an arbitrary origin) that locates the volume elements of size $\mathrm{d}\mathbf{R}$ (Bohren and Huffman, 1983). The RGA form factor only depends on the particle shape, the scattering direction and the scattering size parameter. One interesting property of the form factor is that it does not depend on the particle mass. This means that it is possible to derive a parametrization for the form factor and the mass of the particles independently from one another.

In the Rayleigh approximation the absorption cross-section is calculated as

$$C_{\mathrm{abs}} = 3kV\mathrm{Im}(K) \tag{5}$$

where the $\mathrm{Im}$ operator denotes the imaginary part.

## 2.2 The Self-Similar Rayleigh-Gans Approximation for particle ensembles

By exploiting the concept of snowflake self-similarity (Westbrook et al., 2004), the Self-Similar Rayleigh-Gans Approximation expands the RGA form factor (Equation 4) into a series of analytic functions. The SSRGA is formulated using five parameters (Hogan et al., 2017, $\alpha_e$, $\kappa$, $\beta$, $\gamma$, $\zeta_1$) that are derived from the ensemble structural properties of snow aggregates.

$$\phi_{\mathrm{SSRGA}}(x) = \frac{\pi^2}{4} \left\{ \cos^2(x) \left[ \left(1 + \frac{\kappa}{3}\right) \left(\frac{1}{2x+\pi} - \frac{1}{2x-\pi}\right) - \kappa \left(\frac{1}{2x+3\pi} - \frac{1}{2x-3\pi}\right) \right]^2 \right.$$
$$\left. + \beta\sin^2(x) \sum_{j=1}^{N_{\mathrm{terms}}} \zeta_j (2j)^{-\gamma} \left[ \left(\frac{1}{2x-2\pi j}\right)^2 + \frac{1}{(2x-2\pi j)^2} \right] \right\} \tag{6}$$

The effective aspect ratio $\alpha_e$ is the ratio between the particle extent along the direction of the propagating wave and the maximum particle extent $D_{\mathrm{max}}$. It should be noted that this property is defined differently from the aspect ratio of the ellipsoid which is best fitting the snow particles and the two quantities can differ substantially (Jiang et al., 2017). $\alpha_e$ is only used to scale the argument of the $\phi_{\mathrm{SSRGA}}$ term by computing the size of the particle along the propagation direction.

$$x = kD = k\alpha_e D_{\mathrm{max}} \tag{7}$$

The average mass distribution of the snow particles is described by the kurtosis parameter $\kappa$. This parameter describes how much the mass distribution deviates from a cosine function ($\kappa$=0, see Hogan and Westbrook, 2014) and partially resembles the definition of the kurtosis as a statistical moment. Positive values of $\kappa$ indicate a more pronounced central peak of the mass distribution whilst negative values are associated with a more uniform distribution of the mass within the snowflakes.



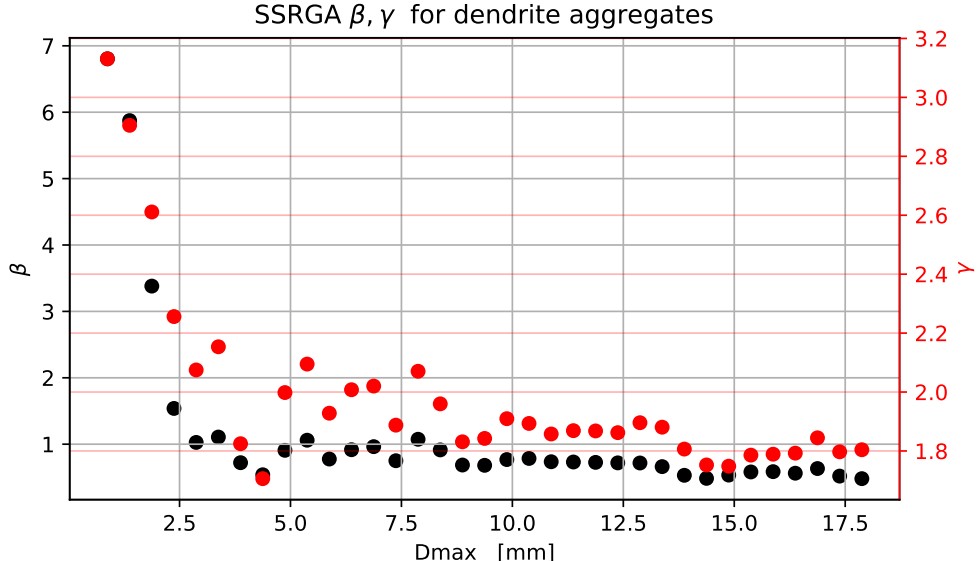

**Figure 1.** Example of the evolution of the fitted SSRGA parameters $\beta$ (black) and $\gamma$ (red) as a function of the snowflake maximum diameter for a aggregates of dendrites. Note that the scales for the two quantities are different and that they are reported at the opposite sides of the graph.

The remaining three parameters $\beta$, $\gamma$ and $\zeta_1$ describe the power spectrum of the mass fluctuations around the average structure. Under the assumption of structural self-similarity, this power spectrum follows closely a power-law relation (Sorensen, 2001). $\beta$ is the prefactor of this power-law and represents the amplitude of the mass fluctuations which decay over the spectrum

wavenumber with a rate of $-\gamma$. The $\zeta_1$ factor is a correction factor that accounts for the deviations that are frequently found for the power spectrum at the first wavenumber from the power-law fit (all the other $\zeta_j$ are equal to 1) (Hogan et al., 2017).

Figure 1 shows the evolution of $\beta$ and $\gamma$ as a function of the snowflake maximum diameter for an ensemble of aggregates of dendrites. The SSRGA parameters are found to change very fast at small diameters and quickly approximate a constant value as the self-similar regime is approached. The theory of snow aggregation (Westbrook et al., 2004) predicts that few aggregation

steps are required to enter the self-similarity regime. This means that once the SSRGA coefficients have stabilized they can be reasonably assumed to be constant and can be used also to calculate the scattering properties of particles larger than the ones used in the ensemble. For small particles the size-resolved fits (Hogan et al., 2017; Leinonen et al., 2018b) provide sets of SSRGA parameters that are valid locally for the range of sizes used in the fitting procedure. For even smaller particles, i.e. sizes at which the snowflakes exists only in terms of single monomers, the assumption of self-similarity is clearly not valid.

However at such small particle sizes the form factor (Eq. 6) reduces to the value of 1 and the SSRGA function reduces to the pure Rayleigh approximation which is not dependent on the particle internal structure.





### 2.2.1 Limitations of SSRGA

As a consequence of the fact that RGA considers the scattering from a particle to be the linear superposition of Rayleigh scattering events, the resulting phase function (Equation 3) exhibits also a Rayleigh-like angular dependency. In particular this means that the two polarimetric components of the scattered field would be equal at forward and backward directions and therefore it would not be possible to study, for example the radar polarimetric properties of particles using RGA.

The porous structure of snowflake particles is assumed to ensure the validity of the first RGA criterion (Equation 1) (Sorensen, 2001). This is because the low effective density of snowflakes leads to a weak electromagnetic interaction among their inner parts. Therefore it is expected that the higher density of a rimed snowflake tends to violate the criterion. The second RGA criterion (Equation 2) indicates, that higher frequencies or larger particles might also break the RGA assumptions. Since SSRGA is based on RGA it is expected to produce the most significant errors for greater densities, larger sizes or higher microwave frequencies. The question of whether the scattering properties of fractal-like particles can be calculated assuming the RGA validity criteria (Equations 1 and 2) has been investigated in previous studies (Farias et al., 1996; Sorensen, 2001; Westbrook et al., 2004). Leinonen et al. (2018a) have further evaluated weather RGA and SSRGA methods are suitable for computing the radar backscattering cross-section for rimed particles and concluded that for not too heavily rimed particles, at frequencies up to 94 GHz, SSRGA can be applied with acceptable accuracy.

## 3 The snowScatt package

The snowScatt tool has been designed to provide a similar interface structure as commonly used scattering databases, such as *scatdb* (Liu, 2008). The additional components, for example to derive individual SSRGA coefficients, are envisioned to help the tool collection of SSRGA parameters to grow while providing the SSRGA coefficients and derived quantities in a consistent manner.

The structure of the snowScatt package is illustrated in Fig. 2. snowScatt is designed to be modular and each component can be used as an independent program. The main database is provided by the snow library which contains the snowflake microphysical properties. Together with the dielectric model for ice, the SSRGA parameters are used by the core SSRGA program to compute the single scattering properties. The mass and area parametrization can be used by the hydrodynamic model component to estimate the terminal fall speed of the snowflakes. Finally, the single scattering and microphysical properties of the snowflakes can be integrated over a particle size distribution (PSD) by the radar simulator to produce idealized synthetic Doppler radar measurements.

### 3.1 The snow library

Although the number of scattering databases of realistically shaped snow particles is constantly increasing, the variability of available particle properties, especially for rimed particles, is still limited (Kneifel et al., 2018). The snowScatt package provides access to the microphysical and scattering properties of an extensive library of snow particle models comprising





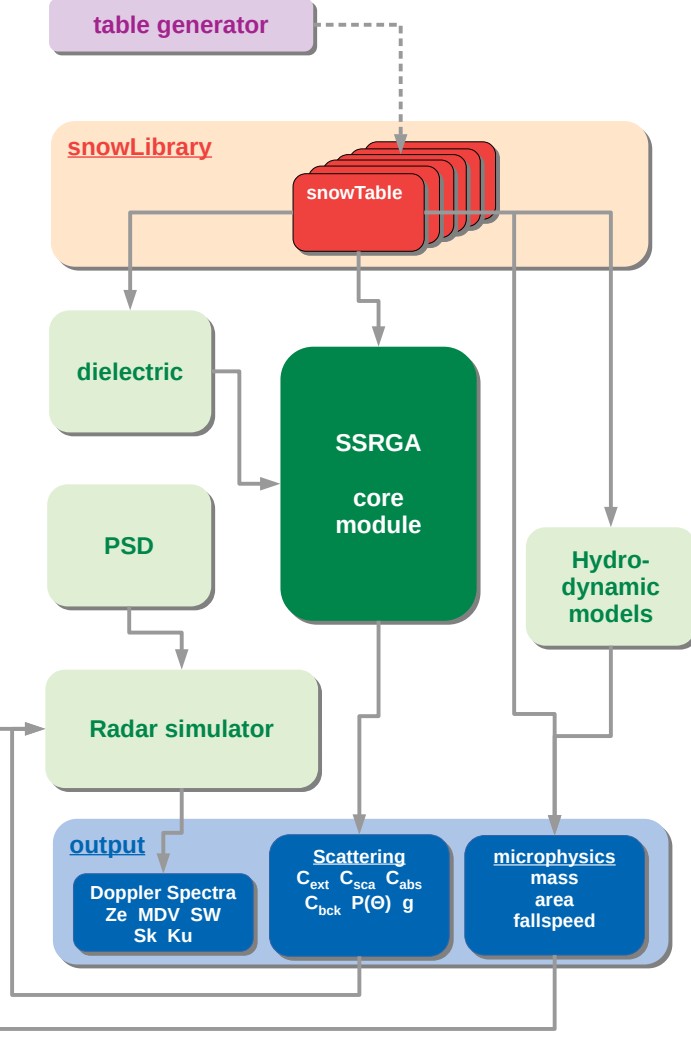

**Figure 2.** Schematics of the modules included in the snowScatt package and basic workflow. The various components are color-coded according to their primary use. The reddish and green blocks identify respectively the data and algorithmic components of the package. The auxiliary "table generator" package, that can be used to extend the snowLibrary database, is colored in violet. Even though the output is not technically a module of the snowScatt package it is here highlighted in blue coloring.





approximately 50 thousand rimed and unrimed aggregates. The aggregate shapes used have been generated in previous studies (Ori et al., 2014; Leinonen and Szyrmer, 2015; Karrer et al., 2020). The details of the aggregation and riming models can be
found in the cited literature.

In general, the particle types included in the snow library can be roughly divided into four classes of snowflakes: rimed aggregates from Leinonen and Szyrmer (2015), aggregates of columns generated by Ori et al. (2014), unrimed aggregates generated using the code described in Leinonen and Moisseev (2015) consisting of a single type of monomers as well as aggregates consisting of a mixture of monomer types (Karrer et al., 2020). Examples of the main aggregates types included in
snowScatt are shown in Fig. 3.

The rimed aggregates included in snowScatt are based on Leinonen and Szyrmer (2015) (hereafter named LS15). There are three riming scenarios available: riming mode A (where riming and aggregation take place simultaneously), riming mode B (the rime ELWP (effective liquid water path) is added to the existing aggregate subsequently) and riming mode C, where the rime ELWP is added to a single ice crystal, in order to recreate conditions of the pure rime growth of graupel-like particles.
The available ELWP, as well as a specific description of the composition (e.g. monomer types used, size of monomers, total amount of aggregates per class) of the aggregates can be found in table 1. For further details on the aggregate properties, the reader is referred to Leinonen and Szyrmer (2015).

As a second class, snowScatt provides the physical and scattering properties of the particles generated in Ori et al. (2014). The Ori et al. (2014) particles consist of differently sized column monomers that are randomly colliding with each other.
The resulting snow aggregates have a fairly rounded overall shape and a density that is comparable to those of heavily rimed snowflakes of the LS15 type, even if they do not simulate the riming process.

The third class of aggregates consists of approximately 30 thousand aggregates, comprised of needle, column, dendrite and plate monomers (hereafter named Cologne aggregate Ensemble, CaE). The aggregates were generated using the aggregation code described in detail in Leinonen and Moisseev (2015). In order to produce a large variety of shapes and sizes, the monomer
number, type and size have been varied. The monomers are sampled according to an inverse exponential size distribution, whose inverse scale parameter has been varied from 0.05 to 9 mm, assuming minimum and maximum monomer sizes of 0.1 and 3 mm. Further details on the structure of the aggregates can be found in table 1.

The fourth aggregate class in the snow library contains aggregates that are made up of a mixture of column and dendrite monomers (CaE-mixture). The mixture aggregates used here are equivalent to the "Mix2" aggregates described in Karrer et al.
(2020), for which the aggregation code from Leinonen and Szyrmer (2015) has been extended to allow the use of a mixture of monomers. The monomers are sampled from an inverse exponential size distribution where the monomers with Dmax < 1 mm are columns, and the ones larger are dendrites. The inverse scale parameter of the size distribution was varied from 0.05 to 3 mm, with minimum and maximum monomer sizes of 0.1 and 3 mm respectively.

The basis for the calculations performed by the snowscatt core are text-files (snow-tables in Fig. 2) which contain the size
resolved SSRGA coefficients as well as physical particle properties such as the mass and cross-sectional area that are described in the form of power-law fits. The particle properties are defined with respect to the snowflake maximum dimension.

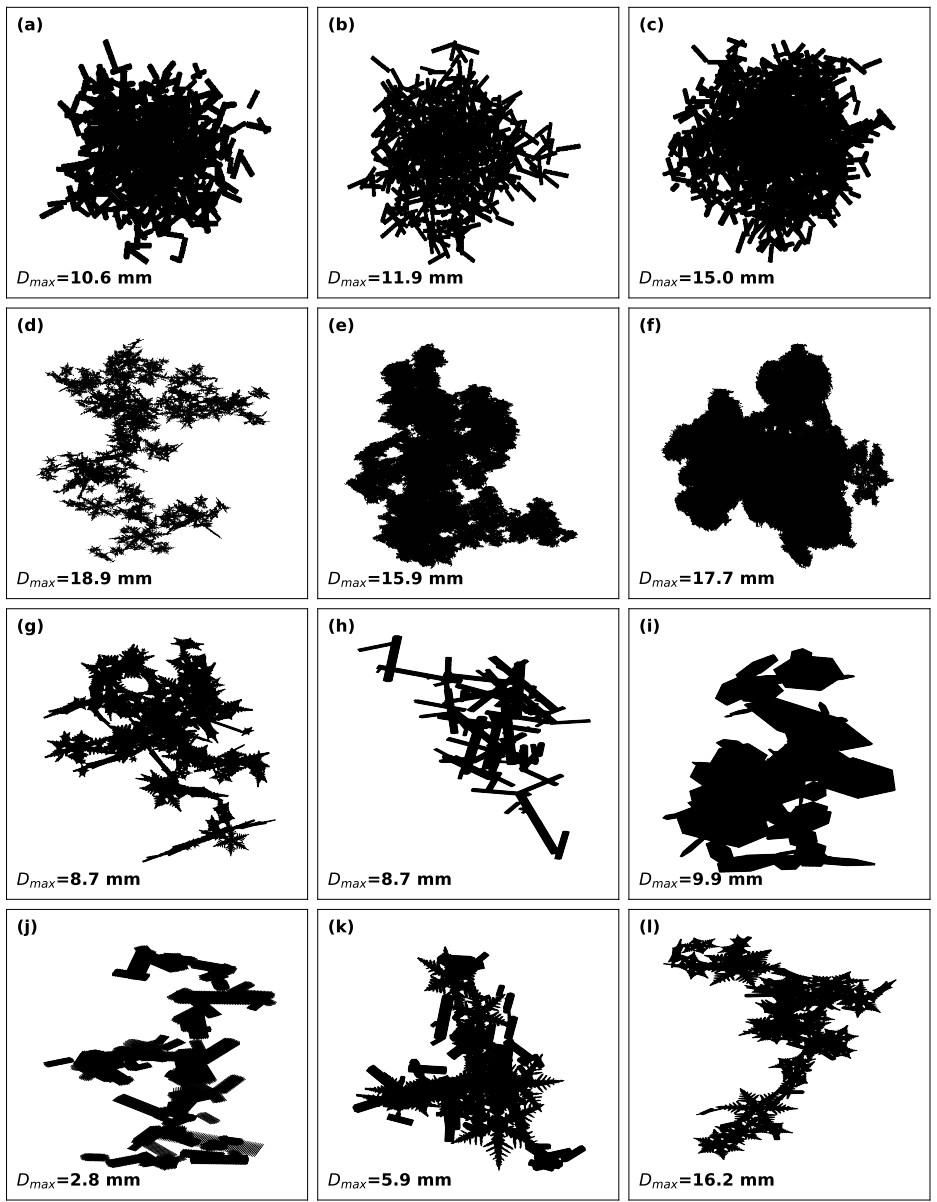

**Figure 3.** Example of the different aggregates available in snowScatt. First row shows aggregates described in Ori et al. (2014). The second row shows the Leinonen and Szyrmer (2015) aggregates, with different degrees of riming: panel (d) is unrimed, panel (e) has $0.5 \ \mathrm{kg \, m^{-2}}$ ELWP (LS15B05) and panel (f) has $1.0 \ \mathrm{kg \, m^{-2}}$ ELWP (LS15B10). The third row shows examples taken from the Cologne aggregate ensemble (CaE), where the aggregate in (g) consists of dendrites, (h) of needles, and (i) of plates. The fourth row gives examples of the CaE-mixture, with different amounts of columns and dendrites present.





**Table 1.** Description of aggregates available in the snowScatt tool. For mD,AD,vD relations see fig. 4

| Aggregate class | ELWP | monomer types | size of monomers | # of aggregates |
|---|---|---|---|---|
| LS15 unrimed | 0.0 | dendrites | 0.1-3 mm | 1270 |
| LS15 A | 0.1 | dendrites | 0.1-3 mm | 1216 |
| | 0.2 | dendrites | 0.1-3 mm | 1268 |
| | 0.5 | dendrites | 0.1-3 mm | 1100 |
| | 1.0 | dendrites | 0.1-3 mm | 1067 |
| | 2.0 | dendrites | 0.1-3 mm | 316 |
| LS15 B | 0.1 | dendrites | 0.1-3 mm | 1260 |
| | 0.2 | dendrites | 0.1-3 mm | 1397 |
| | 0.5 | dendrites | 0.1-3 mm | 1219 |
| | 1.0 | dendrites | 0.1-3 mm | 713 |
| | 2.0 | dendrites | 0.1-3 mm | 379 |
| LS15 C | rime growth | / | / | 1145 |
| Ori14 | 0.0 | columns | 0.1-3 mm | 807 |
| CaE | 0.0 | needles | 0.1-3 mm | 7480 |
| | 0.0 | columns | 0.1-3 mm | 7480 |
| | 0.0 | plates | 0.1-3 mm | 7480 |
| | 0.0 | dendrites | 0.1-3 mm | 7480 |
| CaE-mixture | 0.0 | columns and dendrites | 0.1-3 mm | 4927 |

For the computation of the scattering and the terminal fall velocity, the code assumes the snow particles to follow the mass-size and area-size fits as derived from the snow-table. Those mass and area fits are capped at small sizes by the maximum theoretical mass and cross-section of a solid sphere of the same size. The default assumption can be overridden by specifying the sets of masses and areas to be used in the internal computations. This possibility is particularly useful, because it allows to use snowScatt to forward simulate the outputs of numerical weather prediction models by ensuring internal consistency with the snow microphysical properties assumed in the model (Mech et al., 2020; Ori et al., 2020a).

### 3.1.1 Extending the snow library

snowScatt also offers the tools required to fit the microphysical and SSRGA parameters from an ensemble of snowflake shapes. The tool produces a table formatted according to the snowScatt internal conventions that can be imported at runtime and immediately used along with the sets of snow particles already included in the snowScatt library. This would provide an easy way to extend the snowScatt library to an even larger ensemble of snow properties.





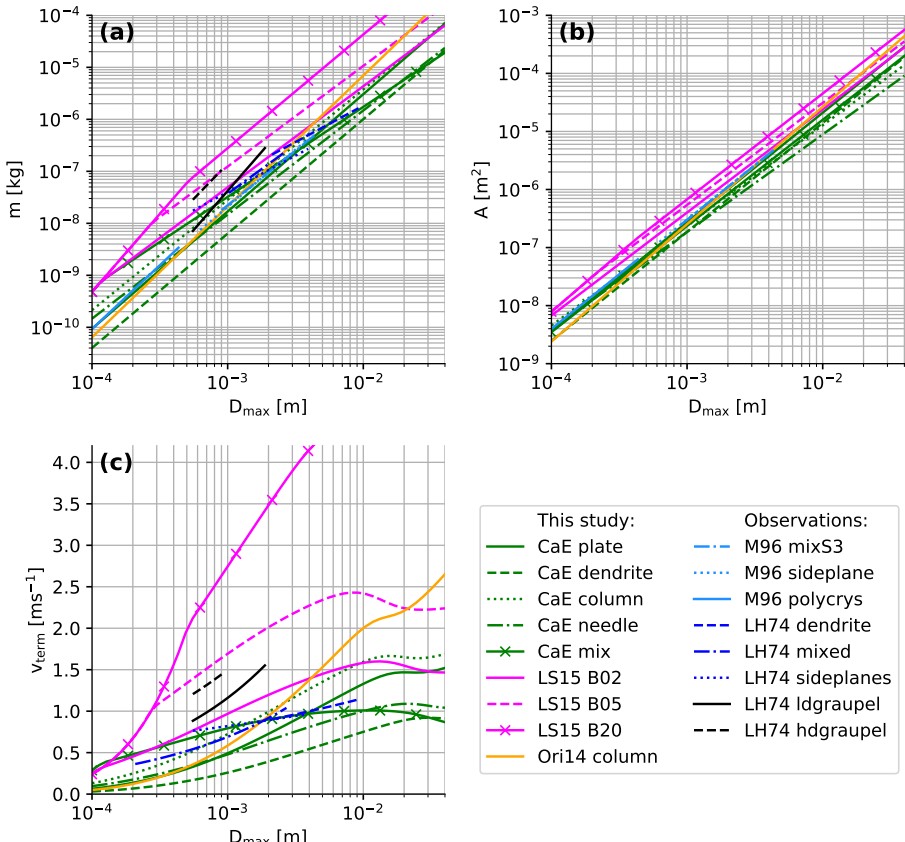

**Figure 4.** Mass a), projected area b) and terminal velocity c), of a selection of the simulated aggregates included in snowScatt and two frequently used in-situ studies (Locatelli and Hobbs, 1974; Mitchell, 1996). Terminal fall velocity in c) is calculated using the hydrodynamic model from Böhm (1992).

### 3.2 Microphysical properties

A comparison of the microphysical properties of a selection of the snowflake models included in snowScatt to relations derived from in-situ observations (Locatelli and Hobbs, 1974; Mitchell, 1996) is presented in Fig. 4.

Both the mass (Fig. 4a) and projected area of the simulated and observed snowflakes (Fig. 4b) follow closely typical power-law relations. The range of the simulated snowflake properties is larger than in the observations, but the observed relations populate in the middle of the simulated ensemble spread. Figure 4 also shows that the size range for which the in-situ observations have been derived is much smaller compared to the simulated particles. The largest difference appears in Fig. 4a for the graupel particles of Locatelli and Hobbs (1974), which show a much steeper slope than the rimed aggregates. This might be





simply due to the fact that the rimed particles presented in Leinonen and Szyrmer (2015) are still less rimed than the graupel observed by Locatelli and Hobbs (1974).

snowScatt implements different hydrodynamic fallspeed models (Böhm, 1992; Khvorostyanov and Curry, 2005; Heymsfield and Westbrook, 2010) with the Böhm (1992) being the default choice as it has been found to most closely match in-situ

observations (Karrer et al., 2020). The fallspeed models calculate the terminal fallspeed of the aggregates by equating the gravitational force (that scales with particle mass) with the drag force (that scales with particle area). The environmental air conditions of pressure, temperature and humidity are also affecting the simulated fallspeed since they change the air viscosity. By default the fallspeed is computed at standard condition (1000 hPa, 15°C), but the user can change this options at runtime. Optionally the Foote and Du Toit (1969) correction for density can also be used to calculate the fallspeeds at non-standard

conditions.

The terminal fall velocities of the snowScatt aggregates computed with the Böhm (1992) model are compared in Fig. 4c with in-situ observations. Again, in-situ observations cover a limited range of sizes. The terminal fall speeds outside of the observed sizes are usually obtained by extrapolating functional relations fitted to the observations. As the particle properties implemented in snowScatt have been calculated with an aggregation model rather than an empirical particle model (Tyynelä

and von Lerber, 2019), the microphysical particle properties are represented for the entire size range in a physically consistent way (Karrer et al., 2020).

### 3.3 Scattering properties

The snow scattering properties are calculated by snowScatt using the SSRGA method. The calculated quantities include the absorption cross-section $C_{abs}$ (Eq. 5), the scattering cross-section $C_{sca}$, the extinction cross-section $C_{ext} = C_{sca} + C_{abs}$, the

radar backscattering cross-section $C_{bck} = \sigma(\pi)$, the asymmetry parameter $g$ and the phase function $P(\theta)$ (Eq. 3). $C_{sca}$ is computed by integrating $\sigma(\theta)$ over the whole solid angle.

$$C_{sca} = \int_0^\pi \sigma(\theta) \sin(\theta) d\theta \tag{8}$$

$$P(\theta) = \frac{\sigma(\theta)}{C_{sca}} \Rightarrow \int_0^\pi P(\theta) \sin(\theta) d\theta = 1 \tag{9}$$

$$g = \frac{\int_0^\pi \sigma(\theta) \sin(\theta) \cos(\theta) d\theta}{\int_0^\pi \sigma(\theta) \sin(\theta) d\theta} \tag{10}$$

The integrals in Eq. 8, 9 and 10 are performed numerically by sampling the $[0, \pi]$ domain with 181 (default value corresponding to 1 degree resolution) equally spaced angles. The number of angular subdivisions also reflects the resolution of the output $P(\theta)$ and can be adjusted individually in snowScatt.

In Fig. 5 we illustrate how our scattering properties generated by the large aggregate ensemble and SSRGA in snowScatt compare to common approximations, such as spherical and spheroidal ice-air mixtures ("soft spheroids"). In addition, the

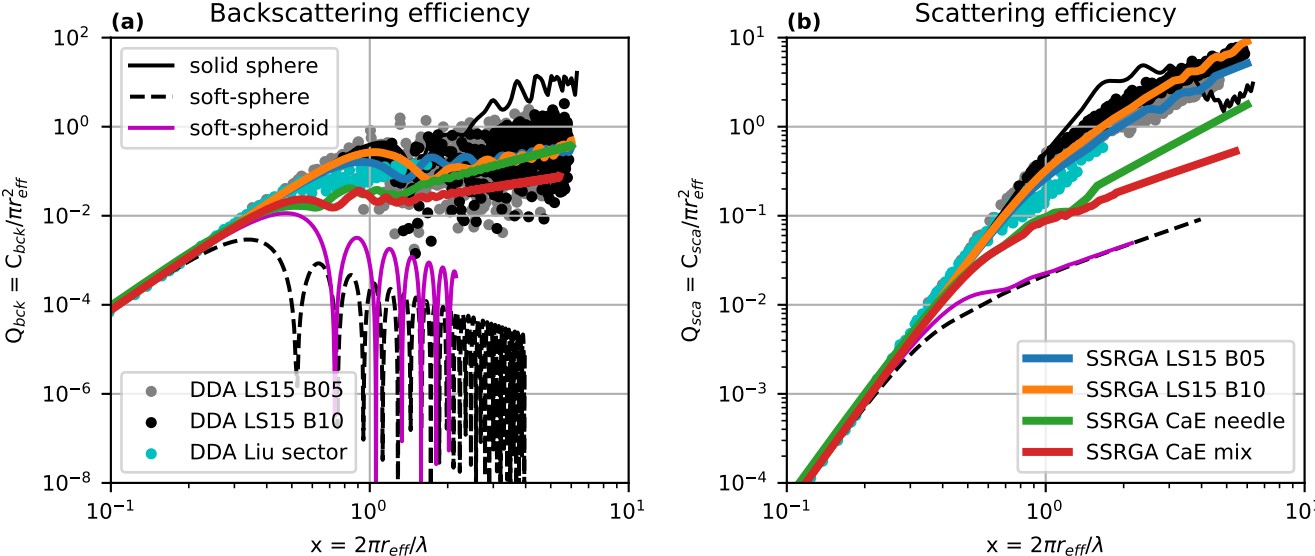

**Figure 5.** Comparison of commonly used scattering methods for snowflakes in terms of backscattering (panel a) and extinction (panel b) efficiencies. Efficiency is defined as $Q = C/\pi r_{\mathrm{eff}}$ with the effective radius $r_{\mathrm{eff}}$ being the radius of an equal mass ice sphere and $C$ the corresponding (backscattering or extinction) cross-section. Scattering efficiencies are plotted against the size parameter defined with respect to the effective radius $x = 2\pi r_{\mathrm{eff}}/\lambda$. This representation allows to combine the dependence of scattering with respect to both the mass of the particle and the electromagnetic wavelength. The soft-sphere and soft-spheroid (aspect ratio equals 0.6) approximations are obtained by assuming a mass-size relation following Brown and Francis (1995) and give the lowest scattering intensity among the presented methods. The solid-sphere generally gives the largest scattering response. DDA results are plotted from published databases for partially-rimed (Leinonen and Szyrmer, 2015) aggregates and the sector snowflake of Liu (2008). The SSRGA computations obtained from some of the unrimed and rimed aggregates included in snowScatt are overplotted for comparison.

SSRGA results are also compared to DDA scattering computations, which are commonly assumed to provide the highest accuracy for complex shapes.

For sufficiently small size parameters (x<0.3), all particles fall into the Rayleigh regime. The size parameter at which the
scattering properties start to exhibit non-Rayleigh effects depends in general on the scattering method and the average density of the particles. A very common feature known from several previous studies are the strong resonances for the spherical and spheroidal models. The "soft" models also exhibit an increasing underestimation of scattering properties which is related to the increasingly lower density and hence lower refractive index of the scattering medium (Petty and Huang, 2010). While the soft spheroids appear to provide the lower limit of the simulated scattering properties, the solid sphere represents the upper limit of
riming with associated highest backscattering and scattering efficiencies.

DDA calculations show that the uncertainty for properties of individual particles is much larger for backscattering than for scattering efficiency. This is particularly true for DDA calculations that assume fixed particle orientations (rimed particles Leinonen and Szyrmer, 2015) than for properties calculated by randomly averaging many snowflake orientations (sector





snowflake Liu, 2008). The more rimed the DDA particles are, the higher is their density and the closer they follow the curve

drawn by solid spheres. The solid sphere model is, in general, a better approximation of the realistically-shaped DDA calculations if compared to the soft-spheroidal models, especially for the most heavily rimed particles. Even if the soft-spheroidal models are able to perfectly match the mass-size relations of the DDA snowflakes their scattering properties are largely deviating from DDA.

The SSRGA results for rimed particles provide a reasonable mean of the DDA derived single scattering properties. Interest-

ingly, the sector snowflakes of the Liu (2008) database is found to populate in between the unrimed and slightly rimed SSRGA curves. We might speculate that the fact that the sector snowflake was found in global assimilation studies to fit the observations best (Geer and Baordo, 2014) hints at the frequent occurrence of slightly rimed aggregates as also found in recent in-situ multi-frequency airborne closure studies (Tridon et al., 2019). Considering the similarity of the Liu sector snowflakes and the SSRGA for moderate rimed particles of ELWP=0.5 $\mathrm{kg\,m^{-2}}$, the SSRGA provides the big advantage of being applicable to any

frequency in the MW, as well as covering a much larger size range.

### 3.4 Dielectric properties

The SSRGA form-factor is, in fact, independent of the refractive index of ice. This makes SSRGA an interesting method to test the sensitivity of snowflake scattering properties with respect to the ice refractive index model or the ice temperature.

The ice refractive index $n$ is thus another input parameter of the SSRGA scattering computation. snowScatt implements

multiple ice permittivity models (i.e. Mätzler, 2006; Warren and Brandt, 2008; Iwabuchi and Yang, 2011) with the Iwabuchi and Yang (2011) being the default choice. The user can specify the ice dielectric model to use in place of the default selection, or override the dielectric model by providing a different value for the complex refractive-index.

An important correction to the computed scattering properties is made through the Rayleigh dielectric factor $K$ (Eq.s 3 and 5). This number is usually computed with the well-known Clausius-Mossotti relation which accounts for spherical objects

and works sufficiently well for isometric shapes. However, the snow aggregates are usually composed of highly elongated or flattened ice crystals. The alignment of the ice mass in the monomers cause an enhancement of the ice polarizability which results in deviations from the pure Rayleigh approximation. In the case of radar reflectivity of single ice crystals those deviations have been found to be up to 4 dB (Hogan et al., 2002).

Exact formulations for the polarizability prescription are available only for a limited set of simplified shapes such as ellip-

soids (Gans, 1912). snowScatt implements the formulation of Westbrook (2014) who has empirically extended the ellipsoid solution to hexagonal ice crystals. This correction is found to improve the matching of the SSRGA-computed scattering properties with those obtained from detailed DDA calculations, but it requires to define a characteristic value for the geometrical aspect ratio of the individual monomers. In snowScatt, this value is estimated from an average among the monomers used for each aggregate type. This procedure is not straightforward in case of monomer mixtures or rimed aggregates. For this cases,

we have estimated the best value of $K$ comparing the SSRGA results with some sample DDA calculations.





## 3.5 Radar Simulator

The snowScatt package also provides a simple built-in radar simulator to compute the radar Doppler spectrum $S_v(v)$ and moments of a distribution of snowflakes. The radar simulator takes as additional input a user defined particle size distribution (PSD, $n(D)$) with predefined functional forms including the modified gamma distribution (Petty and Huang, 2011), the nor-
malized gamma distribution (Testud et al., 2001) and their special case of the inverse-exponential distribution. Based on the PSD and the selected particle type, snowScatt computes the reflectivity size spectrum $S_D(D)$ and the snow fallspeeds $v(D)$. The size spectrum is converted into a velocity spectrum by means of a numerical differentiation of the computed velocity-size relation.

$$\S_D(D) = 10^{18} \sigma_{\mathrm{B}}(D) \, n(D) \frac{\lambda^4}{\pi^5 |K_w|^2}$$
$$S_v(v) = S_D(D) \frac{\partial D}{\partial v} \tag{11}$$

Unlike comprehensive radar simulators (Kollias et al., 2011; Oue et al., 2020; Mech et al., 2020), the Doppler spectrum simulator does not take into account radar instrument specifications or dynamical effects, such as instrument noise level, broadening of the spectrum due to air turbulence, finite beam width, wind shear, etc. The five moments (radar reflectivity $Z_e$, the mean Doppler velocity, the spectrum width, the Skewness, and the Kurtosis) of the Doppler spectrum are computed
directly from the idealized spectrum. Nevertheless, the simulated moments (especially the lower moments such as Ze and mean Doppler velocity) can be compared to real observations when keeping in mind that effects, such as specific radar sensitivity, are not taken into account in our basic radar simulator.

## 4 Evaluation of SSRGA scattering properties

### 4.1 Limits of applicability of SSRGA

As mentioned in Sec. 2.2.1, the assumptions of the SSRGA are expected to become increasingly invalid at higher frequencies and more dense particles. Hogan et al. (2017) demonstrated that the scattering properties obtained from SSRGA match DDA calculations very well up to a frequency of 183 GHz. Leinonen et al. (2018a) evaluated the SSRGA results for rimed aggregates and typical radar bands (Ku, Ka, W-band) and found less than 1 dB bias in backscattering cross-section except for their strongest rimed particles. Considering new radar developments operating in the G-band (Battaglia and Kollias, 2019) as well as upcoming
passive sub-mm satellites (Accadia et al., 2020), there is a need to further test the applicability range of the SSRGA. Only very recently, DDA databases are available covering frequencies up to the sub-mm range and a range of realistically shaped particles (874 GHz, Brath et al., 2020). If reliably applicable, the ensemble scattering properties of the SSRGA would be a valuable complement of these databases as DDA simulations especially for large aggregates at high frequencies become extremely computationally expensive and therefore only very few of them are included.




The sets of particles included in our comparison comprise 48 different aggregate shapes including two types of unrimed and two rimed aggregates. The rimed aggregates of dendrites are taken from the Leinonen and Szyrmer (2015) database and are generated assuming an equivalent liquid water path of 0.5 and 1.0 $\mathrm{kg\,m^{-2}}$. The unrimed particles are aggregates of needles and aggregates of mixed column and dendrite crystals (generated with the procedure introduced in Karrer et al. (2020)). All 48 snowflakes have sizes ranging from 4 to 20 $\mathrm{mm}$. The frequency range investigated goes from radar S-band (1.8 GHz) to the

highest frequency (874 GHz) including all the radiometric channel of the International Submillimetre Airborne Radiometer (ISMAR Fox et al., 2017).

As the SSRGA scattering properties represent the scattering properties of an ensemble of particles, a direct evaluation with DDA would require DDA calculation for each ensemble member or at least a representative number. Considering the large frequency and size range for which we aim to test the SSRGA, this approach is unfeasible due to the extremely high

computational resources necessary. In order to approach the single particle scattering properties best, we applied the individual mass and size of the particle used for DDA to the SSRGA instead of the ensemble-averaged mass and size. Even though the individual differences of single particle scattering properties from SSRGA might not be too meaningful, we are most interested in any bias appearing at a certain combination of frequency and degree of riming.

The reference scattering method used in the present study is the DDA (Draine and Flatau, 1994). The accuracy of DDA is

considered to be limited by the size $d$ of those individual dipoles. According to a recent evaluation (Zubko et al., 2010) the criterion $|n|kd < 0.5$ is sufficient to ensure reliable DDA results. The aggregates used in this comparison are represented by volume elements whose size is 10 $\mathrm{\mu m}$. This discretization level is fine enough to validate the $|n|kd$ criterion for ice particles up to a frequency of 1.35 THz. On the other hand, the rimed aggregates used in Leinonen and Szyrmer (2015) are defined using a dipole resolution of 40 $\mathrm{\mu m}$ which would satisfy the $|n|kd$ criterion only up to 340 GHz. In order to avoid the coarser

definition of the rimed aggregates to introduce accuracy issues in the DDA computations, the resolution of those shapes have been artificially increased by means of a dipole-splitting method (Ori and Kneifel, 2018). This methodology keeps the shape of the particle intact while it increases the dipole resolution to meet the DDA accuracy standards. The DDA implementation we used in this work was ADDA (Yurkin and Hoekstra, 2011). The refractive index model used for both DDA and SSRGA computations is the one of Iwabuchi and Yang (2011) at a reference temperature of 270 K. The DDA results are obtained for a

vertical incident direction of and averaged over a large number of particle orientations (azimuth).

The comparison of scattering, absorption, and backscattering cross-sections (Fig. 6a-c) obtained by SSRGA and DDA show a remarkably good match even up to 874 GHz. Most surprising is the very low bias found for backscattering (Fig. 6c) over the entire frequency range for unrimed as well as rimed particles. The increasing scatter (up to factor of 4 corresponding to 6 dB) of the single particle backscatter with higher frequency and more riming is not surprising as it depends strongly on

the morphology of the individual snowflake. In fact, they reflect the variability of the single particle backscattering properties around the ensemble average. Fortunately, the deviations are uniformly distributed to positive and negative values and there is no evident bias or drift of one model with respect to the other. The low overall bias implies that the SSRGA can indeed be safely applied to radar applications even up to very high frequencies (beyond G-band).



**Figure 6.** Comparison of microwave absorption and scattering properties of single snowflakes calculated with SSRGA and DDA methods. The compared quantities are the total scattering cross-section $C_{sca}$ (panel a), the absorption cross-section $C_{abs}$ (panel b), the radar backscattering cross-section $C_{bck}$ (panel c) and the asymmetry parameter $g$ (panel d). The frequency color-scales (highest frequency included is 874 GHz) are separated for rimed and unrimed particles to facilitates distinguishing between the two particle categories. The unrimed particles are aggregates of needles and aggregates of mixtures of columns and dendrites with sizes ranging from 4 to 20 mm. The rimed particles are dendrite aggregates taken from Leinonen and Szyrmer (2015) database, mode B (subsequent aggregation and riming) assuming an equivalent liquid water path (ELWP) of 0.5 and 1.0 kg m$^{-2}$. On panels a, b and c two dotted lines mark the $\pm$ 3 dB range of variation from the perfect match.





The overall best match for DDA and SSRGA is found for absorption cross-section (Fig. 6b). The discrepancies between the two models for this quantity are always within 60%. Also for the scattering cross-section (Fig. 6a) the SSRGA matches the DDA calculations surprisingly well. For scattering cross-sections exceeding $10^{-5}$ m$^2$ the SSRGA starts to increasingly overestimate the DDA values. This scattering cross-sections are typically reached for unrimed aggregates sizes in the range of 10 mm and frequencies larger than 600 GHz. Nonetheless, for the set of unrimed particles used in this experiment the maximum deviation was found to be in the order of 4 dB. For rimed particles, the deviations start as expected at lower sizes and frequencies. Moderately rimed aggregates (ELWP=0.5 kg m$^{-2}$) of 10 mm can be safely applied up to 325 GHz. The SSRGA for same sized aggregates but with heavy riming (ELWP=1.0 kg m$^{-2}$) starts exhibiting deviations at frequencies larger than 220 GHz.

The asymmetry factor $g$ (Fig. 6d) shows more significant and consistent deviations indicating that SSRGA is frequently overestimating forward scattering with respect to the DDA solution. Interestingly the rimed particles seem to give generally results which are closer to DDA for frequencies lower than 448 GHz. This discrepancy in forward versus backward scattering found for SSRGA should be investigated in future studies as it seems to indicate a need to improve the formulation of the phase function.

## 4.2  SSRGA as ensemble scattering model

As mentioned before, due to the high computational cost of scattering calculations, DDA databases comprise only a limited number of particles. Each single particle included in the database is thus considered to be representative of all the snowflakes of similar size or mass (Geer and Baordo, 2014). The distinctive feature of SSRGA is the characterization of particles' structures with the properties of the ensemble. This is particularly significant in the forward modeling of precipitation because remote sensing applications never look at the properties of single particles, but rather at a very large number of objects with similar characteristics. Figure 5 has shown that the scattering properties of single aggregates with similar sizes span over some orders of magnitude. This means that by characterizing the snowflake single scattering properties with only a few particles, we might introduce uncertainties related to the sub-sampling of the natural variability of snowflake properties.

In order to evaluate this effect, we used the Leinonen and Szyrmer (2015) database for unrimed aggregates of dendrites as a representative ensemble for the natural variability of snow properties. Although the number of particles included in the database is not nearly comparable to the number affecting the measurements of a radar or a radiometer, the used database is one of the DDA databases comprising the largest number of particles (550) of the same type and covering a vast range of particle sizes (from 0.1 to 21 mm). Other DDA databases (Liu, 2008; Ori et al., 2014; Brath et al., 2020) rather focus on providing a wide range of parameters such as frequency and temperatures, which restricts the amount and sizes of particles implemented due to limited computational resources. Instead, the Leinonen and Szyrmer (2015) database provides the scattering properties only at few radar frequencies, fixed orientation and does not account for temperature variability.

We have divided the DB into size bins that are 1.5 mm large. The resulting distribution is quite uniform with about 35 aggregates per bin size (Fig. 7a). We assume that the average properties of those 35 aggregates are representative of the properties of the entire snow population for each size bin. The resulting average radar backscattering cross-section at Ka-

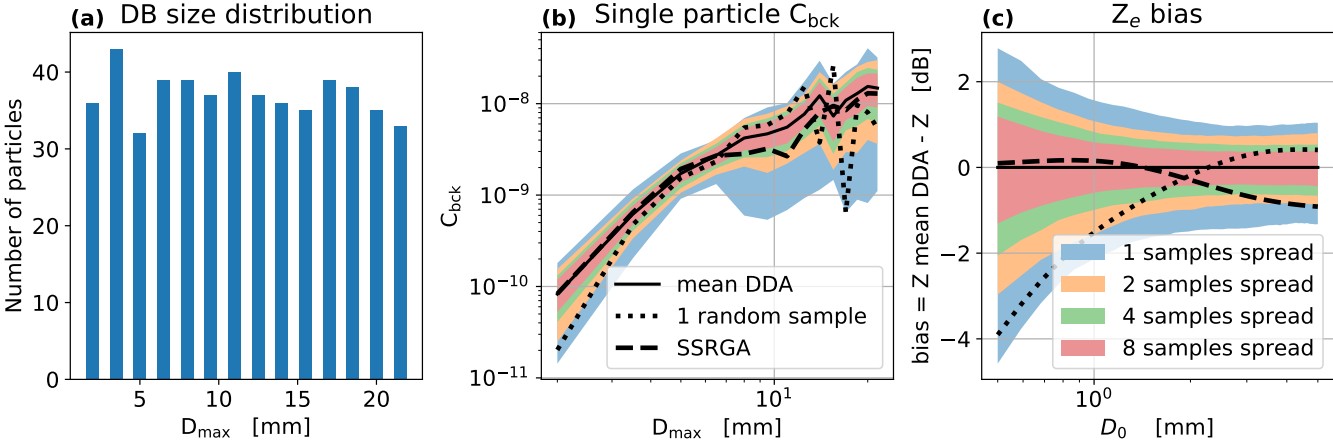

**Figure 7.** The uncertainty introduced by sampling the natural variability of snow properties using only few particles to represent the whole population of snowflakes. Panel a shows the size distribution of snowflakes included in the reference database. Panel b shows the 35.6 GHz backscattering cross-section as a function of size for the reference database (continuous line), the SSRGA solution (dashed line) and a random sample of 1 particle per size bin in the DDA database. The colored areas show the 10-90 percentile variability caused by randomly sampling the DDA database with 1, 2, 4 or 8 particles per size bin. In panel c the bias in the forward simulated radar reflectivity as a function of the characteristic size $D_0$ of the assumed inverse exponential size distribution is shown.

band (35.6 GHz) is shown in Fig. 7b showing a remarkable agreement between the SSRGA and the average DDA solution. The dotted line in the plot represents the backscattering properties given by a random sample of only 1 particle for each
size bin, simulating the effect of sub-sampling in DDA databases. In order to evaluate the uncertainty related to this random process, we have repeated the sampling experiment 1000 times and computed the 10th to 90th percentile spread of the resulting backscattering properties (blue area in Fig. 7b). It is shown that the uncertainty associated with this drastic sub-sampling can be as high as a factor of 10. We have repeated the experiment allowing a progressively better representation of the database variability by picking 2, 4 or 8 snowflakes per size bin. The corresponding uncertainty areas show that 8 samples per size bin
(100 particles in total) are needed to match the uncertainty level of the SSRGA ensemble computations.

In order to evaluate the effect of this uncertainty to radar applications we have integrated the resulting backscattering cross-sections over various inverse exponential PSDs of the form $N(D) = N_0 \exp(-D/D_0)$. The computed reflectivity bias (difference with respect to the reference DDA average in dB) is plotted in Fig. 7c as a function of the PSD characteristic size $D_0$. The bias between the two reflectivities is obviously not depending on the PSD concentration parameter $N_0$. The SSRGA bias
remains within 1 dB for the entire range of simulated $D_0$. In the case of DDA, the spread of uncertainty in the worst case scenario of only 1 sample per size bin goes up to 4 dB for the smallest value of $D_0$ and reduces to 1.5 dB for the largest. Larger $D_0$ are associated with broader PSDs and thus a greater variability of particle properties across the PSD. This partially compensates for the reduced representativeness within each size bin.





The limited number of particles included in the reference database is the major limitation to a proper evaluation of the
potential bias caused by the insufficient representation of the variability of snowflake properties in scattering databases. This
would require a much larger database that could comprise thousands of particles per size bin. This idealized experiment is meant
to provide an indication of the importance of snowflake sub-sampling to use-case scenarios, such as the forward modeling of
radar reflectivity based on a database with a low number of specific particle shapes.

Another potential source of uncertainty given by limited DDA databases is the size of the largest snow particle. SSRGA gives
a physical way to extrapolate the scattering properties of aggregates to particles of any size, while DDA can only extrapolate
from a best-fit curve. Given the high non-linearity of the scattering processes this could cause large uncertainty in the modeling
of scattering properties of very large snowflakes. This effect was not evaluated in the presented experiment since the integration
over the PSD was always truncated at the size of the largest available DDA snowflake.

## 5  Application example

New model developments, such as the P3 scheme (Morrison and Milbrandt, 2015) or Lagrangian super-particle models (Brdar
and Seifert, 2018; Shima et al., 2020) pose major challenges to current forward operators as their hydrometeor properties (e.g.,
mass - size relation) are not fixed but explicitly predicted by the model. Current DDA databases are only of limited use for such
simulations as their particle properties are fixed. Similar to T-Matrix calculations, the SSRGA provides for those applications
the necessary flexibility while still providing sufficiently accurate scattering properties. In order to demonstrate the SSRGA
capabilities in this respect, we apply the snowScatt to output from the P3 scheme (Morrison and Milbrandt, 2015) implemented
in the ICOsahedral Non-hydrostatic model (ICON Heinze et al., 2017).

Usually, bulk microphysical schemes represent the ice variability using multiple categories (e.g. Seifert and Beheng, 2006).
The peculiarity of the P3 scheme is to substitute the ice-phase multi-category architecture with a single category defined by
continuously variable properties. In particular, the scheme predicts the evolution of two prognostic variables, namely the rime
fraction and bulk rime density that defines how much of the PSD is affected by riming and the intensity of the riming degree.
Basically, these two quantities define the range (minimum and maximum size) of the rimed snowflakes and their mass-size
relation. The test scene used in this example is the output of an ICON (Dipankar et al., 2015) large eddy simulation coupled[1]
with the P3 cloud microphysical scheme for the 24 November 2015. In this study, the so-called "Meteogram" output is used,
which consists of the time evolution of the cloud field closest to the JOYCE (Jülich ObservatorY for Cloud Evolution, Germany)
measurement facility (Löhnert et al., 2015). Model output is provided every 9 s.

We used the mass-size relations, derived from the model output, to define the scattering properties of the snowflakes used in
the forward simulations. The range of rimed particles included in snowScatt are characterized by the ELWP used to simulate
the riming process (Leinonen and Szyrmer, 2015). While convenient for modelling riming on a single particle level, ELWP

---

[1]A paper describing the implementation of the P3 cloud microphysical scheme in the ICON model is under preparation (Tontilla, Karrer, Milbrandt, Morrison, Ori, Kneifel and Hoose "The predicted particles properties (P3) cloud microphysics scheme in ICON-LEM: Idealized tests and application to HOPE cases")





is not a quantity readily available from the P3 scheme and thus it is not straightforward which particle scattering to use. We
decided to empirically relate ELWP with the mass-size relationship predicted by the P3 scheme. This connection is facilitated
by the fact that all m-D power-laws fitted to the modeled rimed particles are characterized by a similarly valued exponent
(Leinonen and Szyrmer, 2015). First, we have derived new best-fit prefactor parameters by fixing the exponent value to 1.9 as
it is assumed by the P3 scheme. The SSRGA set used for the forward modeling of rimed snow is then assumed to be the one
whose mass-size prefactor most closely matches the one given by the P3 model. Unfortunately, the particular case used in this
study exhibit a limited amount of riming and the resulting set of parameters used in the forward simulations was always the one
with ELWP=0.1 $\mathrm{kg\,m^{-2}}$. Therefore, it can be expected that for a case with more intense riming in the model, the improvements
due to using SSRGA will be even more pronounced.

The results obtained from snowScatt are compared to those calculated using state-of-the-art T-matrix and DDA solutions.
Regarding the T-matrix methodology (Waterman, 1965; Mishchenko et al., 1996) we modeled the hydrometeors using oblate
spheroids with an aspect ratio of 0.6 (Matrosov et al., 2005) composed of an homogeneous mixture of air and ice. Each
spheroid matched the maximum size and mass predicted by the model output. The dielectric mixture has been calculated using
the Bruggeman (Bruggeman, 1935) relation which is symmetric with respect to the definition of ice and air inclusions and
allows to continuously span the whole range of hydrometeors densities.

The DDA solution has been calculated by assuming the Liu (2008) sector snowflake (particle id number 9). As mentioned
before, the particles in the Liu (2008) DDA database have fixed particle properties. By simply applying them to the P3 output
would immediately cause inconsistencies in the scattering due to the differences in particle mass for a certain size. In order to
avoid this inconsistency, the backscattering cross-section of the DDA database $\sigma_{\mathrm{DDA}}(D)$ has been scaled for each snowflake
size according to the squared mass predicted by the P3 scheme $m_{\mathrm{P3}}^2(D)$, such that

$$\sigma_{\mathrm{P3}}(D) = \frac{m_{\mathrm{P3}}^2(D)}{m_{\mathrm{DDA}}^2(D)}\sigma_{\mathrm{DDA}}(D) \tag{12}$$

where $m_{\mathrm{DDA}}(D)$ is the mass of the particle included in the DDA database with a maximum size of $D$ and $\sigma_{\mathrm{P3}}(D)$ is the
resulting radar backscattering cross-section used for the forward simulation.

The scattering properties of the liquid hydrometeors (cloud water and rain) have been computed using T-matrix with identical
settings for all forward simulation experiments. Panels a, c and e of Fig. 8 show the W-band (94 GHz) radar reflectivities
computed with the three different scattering methods for snow particles. In the upper parts of the cloud, the particles are
generally small and thus scatter predominantly in the Rayleigh regime. Any inconsistency in the mass-size relation between
P3 and the scattering models would directly manifest itself in deviations of radar reflectivity as the scattering is determined
in the Rayleigh regime solely by $m^2$. Due to the adjustments of masses for the Liu sector snowflakes (Fig. 8e), no mass-
related differences in Ze are found. This consistency is further highlighted in panels d and f of Fig. 8 that show the reflectivity
difference of the T-matrix and DDA solutions with respect to the SSRGA method.

The differences among the reflectivities computed with the three methods become more significant for higher reflectivity
values. The pattern of the differences correlates well with the bulk mean mass of the snowflakes (Fig. 8b) which is here
computed as $<m>=q/N$, where $q$ is the snow mixing ratio and $N$ is the snow particle number concentration. This is



**Figure 8.** Simulated W-band (94 GHz) radar reflectivity on the 24 November 2015 precipitation event over JOYCE based on the output of the ICON model implementing the P3 cloud microphysical scheme. Panels a, c and e show the radar reflectivity forward simulated using respectively the SSRGA, T-matrix and mass-scaled DDA (Liu sector snowflake) to represent the scattering properties of snow. Panels d and f show the reflectivity difference of correspondingly the T-matrix and the DDA solutions with the respect to the SSRGA. Panel b shows the mean snow mass field as predicted by the ICON model.





expected, considering the results of Fig. 5a that shows the largest differences among the single scattering radar cross-sections for the particles with the highest size-parameter. In these situations, the T-matrix methodology shows a clear underestimation of the scattering intensity as it is expected due to the very low density assumed for the largest particles. On the contrary, the

DDA solution shows a general overestimation of the large snowflake reflectivity. This is because the effect of the higher density of the specific particle used (Liu, 2008, sector snowflake) is only partially compensated by the adjustment of the particle mass for larger particles which transition out of the Rayleigh regime (Eq. 12).

The presented application experiment is not meant to prove the better accuracy of one scattering method over another. Although, radar observations for JOYCE are available for this day (Dias Neto et al., 2019), it would be difficult to judge

whether the agreement of the observations and the forward simulations are due to the quality of the spatio-temporal matching of the ICON simulations or the choice of scattering method. However, we might assume that the range of simulated and real hydrometeor contents and related PSDs for the entire case should be similar if the scattering method is appropriate. Therefore, we show in Fig. 9 the triple-frequency (X-, Ka- and W-band) radar measurements recorded during the test case (Dias Neto et al., 2019). The data used for the distributions have been restricted to those associated with temperatures below -5 °C in order

to isolate the signature of frozen hydrometeors. The triple-frequency signatures simulated using the three different scattering methods (SSRGA, T-matrix and DDA sector snowflake) and the ICON output are over-plotted for comparison. Clearly, the SSRGA matches the mean of the distribution very well, while the T-Matrix tends to overestimate and the Liu sector snowflake tends to underestimate the mean especially at larger DWRs related to larger aggregates.

This application example clearly revealed that SSRGA can be used effectively to compute the synthetic unpolarized re-

flectivities of the P3 scheme. SSRGA combines the realism of scattering properties similar to DDA with the flexibility in the computation of snowflakes with different microphysical properties which is characteristic of T-matrix. The applicability of SSRGA is not limited to P3, but it is useful also for the forward simulation of other shape-adaptive ice schemes (Harrington et al., 2013; Jensen et al., 2017; Brdar and Seifert, 2018; Tsai and Chen, 2020). Having a large enough database of snowflake shapes, one can imagine to parametrize the SSRGA coefficients according to the quantities explicitly modeled by these shape-

predictive schemes (such as riming degree, aspect ratio or monomer type) and directly connect the model output with the SSRGA forward simulation.

## 6 Conclusions

With this contribution we aimed at serving the snow scientific community with snowScatt, an innovative tool to access the microphysical and scattering properties of an ensemble of 50 thousand snowflake aggregates. snowScatt makes use of snow

particle shape models in order to calculate the microphysical and scattering properties of snow. The combined derivation of snow microphysical and scattering properties ensures the physical consistency of the modeled quantities. This consistency is a necessary feature in order to properly connect the microphysical properties assumed in weather prediction models with the scattering quantities measured by remote sensing instruments. Moreover, snowScatt enables the study of the snow par-





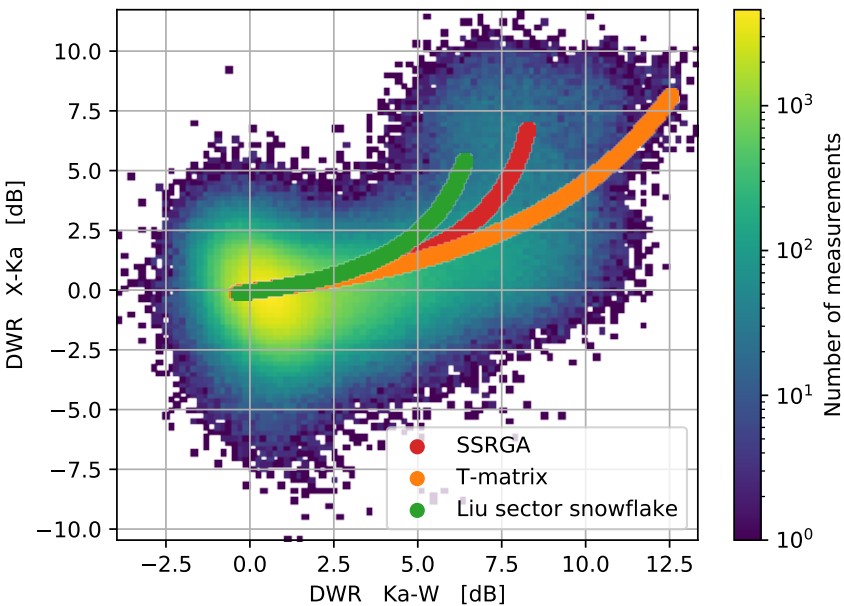

**Figure 9.** 2d-histogram of DWR X-Ka and Ka-W measured on during the TRIPEx campaign (Dias Neto et al., 2019) at the JOYCE supersite on 24 November 2015. The three colored curves show the simulated DWR-DWR relations based on the ICON-P3 model output and the three scattering methods used (SSRGA, T-matrix and DDA).

ticle response to changes in growth processes (e.g. the aggregation and riming simulated by the shape models) from both a

microphysical and scattering perspective.

The current version of the tool provides the properties of different snowflake types including rimed particles and aggregates of different monomer composition. The tool can be easily interfaced with existing forward modeling software and can be extended to include even more particle properties either derived from in-situ observations or other aggregation models.

The scattering properties derived with the SSRGA techniques compare well with DDA reference computations. Despite

the difficulties in the evaluation of ensemble SSRGA results for single particles, the model is proven to be fairly reliable up to 878 GHz, exhibiting minor deviations with respect to the reference DDA calculations only in the case of heavily rimed particles at frequencies higher than 220 GHz. The SSRGA properties are derived for particle ensembles which is advantageous in practical applications. The ensemble properties inherently avoid the sub-sampling problem that affects computationally costly DDA calculations. Moreover, the characteristic scaling of snowflake micro-structure allows SSRGA to extrapolate the

scattering properties to sizes that are even larger than the maximum size included in the shape database.

The set of SSRGA coefficients are not depending on the electromagnetic frequency or the ice refractive index. This makes SSRGA a perfect tool to make sensitivity tests of the snowflake scattering properties with respect to different refractive index



models, temperature regimes or frequency. On the other hand, one must acknowledge that the polarimetric pattern of the SSRGA scattering properties can only follow the Rayleigh form which prevents any application to e.g. radar polarimetry.

One of the main advantages of SSRGA and the large snowScatt library is its flexibility with respect to applications that require continuously changing definitions of snow properties. This feature is tested by forward modeling the radar reflectivity of a P3 test scene. In order to be consistent with the P3 model output, the scattering method needs to be flexible regarding the definition of the snowflake density. snowScatt provides the same level of flexibility as T-matrix while the computed scattering properties reach the level of accuracy of DDA calculations.

The flexibility of the snowScatt methodology is not limited to the forward simulation of the P3 scheme. Provided a reasonable basis of snowflake shapes it is possible to parametrize the SSRGA parameters with respect to any structural property of snowflakes (e.g. rimed fraction, aspect ratio, monomer types). Therefore snowScatt can be used for the forward simulation of complex microphysical schemes that explicitly predict the evolution of snow characteristics.

*Code and data availability.*    This paper presents the snowScatt software toolkit publicly available https://github.com/OPTIMICe-team/snowScatt

and released under the terms of the GNU Public License version 3. The exact version used in the manuscript is archived on Zenodo (Ori et al., 2020b). All codes needed to reproduce the results presented in this paper are included in the examples folder of the snowScatt repository. The necessary input data for the reproduction of the presented analysis are archived also under Zenodo (Ori et al., 2020c)

*Author contributions.*    DO is the main developer and maintainer of the snowScatt package. DO additionally performed the DDA single-scattering computations, the forward modeling of the ICON P3 scene and the various analysis presented in the study. LvT developed the

Cologne aggregate ensemble with significant contributions from MK. LvT also computed SSRGA coefficients and produced the tables included in the snowScatt library. MK modeled and analyzed the microphysical properties of the snow aggregates and significantly contributed to the forward modeling of the P3 test scene. SK initiated the project of deriving SSRGA coefficients from various aggregate models, provided early implementations of the SSRGA algorithm and was heavily involved in the interpretation and testing of the snowScatt results. DO prepared the manuscript with the contributions from all co-authors.

*Competing interests.*    The authors declare no competing interests

*Acknowledgements.*    Contributions by D. Ori, M. Karrer and S. Kneifel were funded by the German Research Foundation (DFG, Deutsche Forschungsgemeinschaft) under grant KN 1112/2-1 as part of the Emmy-Noether Group OPTIMIce. Work provided by L. von Terzi has been supported by the DFG Priority Program SPP2115 "Fusion of Radar Polarimetry and Numerical Atmospheric Modelling Towards an Improved Understanding of Cloud and Precipitation Processes" (PROM) under grant PROM-IMPRINT (Project Number 408011764). We thank the

computing center of the University of Cologne (RRZK) for providing the CPU time on the DFG-funded high-performance computational



cluster CHEOPS. The authors acknowledge the original work of Robin Hogan and Chris Westbrook that developed the SSRGA methodology. We thank Jussi Leinonen for making the snowflake aggregation and riming model publicly available which allowed for the construction of the Cologne aggregate Ensemble. We are grateful to Corinna Hoose and Juha Tontilla for providing the ICON-P3 output that enabled the analysis included in Sect. 5.



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
