# Peer review of "snowScatt 1.0: Consistent model of microphysical and scattering properties of rimed and unrimed snowflakes based on the self-similar Rayleigh-Gans Approximation"

_Geoscientific Model Development, 2020_

## Referee Comment (RC1) · Grant Petty (Referee) · 12 Dec 2020

This paper describes a new modeling system for computing the radiative properties of snow particles in the microwave band. As such, it has the potential to move the field of microwave radiative transfer from its earlier (though still relatively recent) numerical experiments and databases for a limited set of frequencies and particle shapes to a practical community resource that appears to be both easy to use and of wide potential applicability. It includes an impressive database of modeled snow aggregates, both rimed and unrimed, and it utilizes a computationally efficient and flexible numerical methodology – the self-similar Rayleigh-Gans Approximation (SSRGA) – for the single-particle scattering calculations. I don't know of any other research group undertaking something comparably ambitious and versatile, and I predict that this will quickly become a go-to tool for radiative transfer calculations and as a foundation for inverse methods related to both passive and active microwave remote sensing.

Except for a few specific instances noted below, the paper is well-written and quite thorough in describing both the methods and the limitations of the tool.

I visited the github repository and found that the software is convenient to download and install, though I haven't tried using it yet. There's a good start on documentation, though some sections appear not to have been written yet.

My overall recommendation is to publish after considering the comments below.

Minor comments:

lines 54, 131: The SSRGA is introduced here with appropriate citations, but for readers who haven't read those other papers yet, an additional sentence or two explaining what "self-similar" means in this context could be helpful.

line 61: Offhand, at least, I don't know what a "Rayleigh distribution of polarimetric components" is, so maybe a slight elaboration would be useful here as well.

line 77: "parametrized" should be "parameterized"

Eq. (3): The RGA yields a symmetric scattering phase function, as shown by this equation. But I believe that for diameters $D$ (where $D$ is the dimension in the direction of the propagating wave) much greater than about $0.1\lambda$, the phase function quickly shifts toward stronger forward scattering owing to consistently constructive interference in the forward direction (irrespective of size) and varying degrees of destructive interference in the backward direction. Since this is mainly a geometric effect, I'm not even sure whether small $|n - 1|$ eliminates that asymmetry, so I'm wondering whether whether Eq. (2) tells the whole story. In other words, a particle with $kD \sim 1$ or greater, should

not, I don't think, conform to Eq. (3) regardless of whether it satisfies Eqs. (1) and (2). If I'm mistaken on this, please disregard this comment, but it would be worth checking and clarifying, if needed.

line 119: Is $V$ the spherical-equivalent volume?

line 148: Fig. 1 is not completely convincing as regards the purported convergence of $\beta$ and $\gamma$. Any curve starts to look flat as it approaches zero on a linear axis. The point might be made more convincingly if a log vertical axis were used in the plot.

line 467: For what it's worth, Petty and Huang (2010) found that neither Bruggeman nor Maxwell-Garnett dielectric mixing formulas gave the best fit to DDA calculations for soft spheres but rather Sihvola (1989) with an exponent of 0.85.

General:

Several references are made to the computational cost of the DDA method. While true, note that Petty and Huang (2010) demonstrated a variation on the method that at least avoids the extremely large memory requirement of DDSCAT in the case of low-density aggregates and effectively allows smaller dense linear systems to be solved rather than very large sparse ones. In other words, I think DDSCAT might not be the ideal benchmark for evaluating the viability of the DDA approach in a resource-limited computational environment. DDA calculations can be run inexpensively on desktop workstations using the alternative approach.

---

## Referee Comment (RC2) · Anonymous Referee #2 · 2 Jan 2021

Summary:

The paper's primary purpose is to present and describe snowScatt 1.0, a toolkit for computing the scattering properties of unrimed and rimed aggregates from microwave to sub-millimeter electromagnetic wavelengths using the Self-Similar Rayleigh Gans Approximation (SSRGA). A secondary purpose of the paper is to illustrate the value of the SSRGA in the first place.

Review:

[Figure]

The paper is certainly relevant to GMD, and the toolkit will be of value to the community in the years to come. Given the primary purpose of the paper, Sections 2 and 3 are most important and must be written perfectly, providing all of the details necessary to understand the code in detail and to apply it without error. In this regard the paper needs more work, and the focus of this review is on these two sections. Because there are no major issues with the approach, the required revisions suggested for the paper are listed as minor in nature, though certainly required in order to maximize impact of the paper.

McCusker, Westbrook, and Tynella have just come out with a paper in QJRMS with results that address some of the issues raised in this paper. The reference for this paper is

An accurate and computationally cheap microwave scattering method for ice aggregates: the Independent Monomer Approximation Karina McCusker, Chris Westbrook, and Jani Tyynelä

By incorporating relevant results from McCusker et al., e.g., problems with forward scattering in the RGA, the authors will strengthen this paper as well.

Finally, all qualitative comparative language should be removed from the paper and replaced with quantitative measures. This will not be hard to do because quantitative measures appear to be readily available where qualitative language is used.

The PDF file returned to the authors contains mark-up using Acrobat Comments that indicates some of these places. The mark-up also contains other issues the authors might consider addressing.

Detailed Comments on Sections 2 and 3:

0) For an algorithm/data product paper such as this one, every parameter in every algorithm and data product must have clearly specified units. Lacking clear, correct, and internally consistent units, things will get confusing in a hurry.

1) Calling sigma on Line 118 an intensity is confusing. Intensity in the SI system has units of W / sr. Sigma on this line seems to have units of mˆ2 / sr based on Eq. (8). (Eq. 8 is an integral over all 4*pi steradians of solid angle and the result is in mˆ2.) So, it is probably better called a differential scattering cross section if it does indeed have units of mˆ2 / sr.

2) Why have the word "Rayleigh" in front of "dielectric factor" on Line 119?

3) The form factor, itself, has nothing to do with deviations from any quantity. As one factor in a bunch of factors, it might be interpreted as creating deviations in the product of other factors, but to present it as such is to hide its own direct physical significance. So, much of the sentence on Line 120 is not helpful in the context of this paper. Lines 121 and 122, together with Eq. 4, make this clear. For example, there is no deviation involved in Eq. 4, just phase changes as a result of location and direction. Simple and clear these lines are.

4) Is phi in Eq. 3 the same thing as phi_RGA in Eq. 4? If so, please give them the same label. If not, explain the difference between the two. If the same thing, phi_RGA should be a function of x and theta, similar to phi in Eq. 3, and as stated on Line 125.

5) Line 128, Eq. (5), and Line 129 are out of context. Probably best to move them to just before Eq. 8.

6) On Line 12 of the Abstract reference is made to "the set of SSRGA coefficients", and these words occur elsewhere throughout the text. What these coefficients represent is never made perfectly clear. Are these the "five parameters (Hogan et al., 2017, alpha_e, kappa, beta, gamma, eta_1)" that occur on Lines 132 and 133? If so, make this perfectly clear and always use the same words (e.g., "coefficients" or "parameters") to describe them. Otherwise, things get confusing in regards to them.

7) The shape and orientation of any particle to which Eq. 4 is applied are both captured by the integral over V, i.e., where the ice volume elements are actually located. Line

125 correctly characterizes this state-of-affairs. From this perspective, Lines 131 and 132 need more clarity because as currently written Eq. 6 has no dependence on theta. Eq. 6 looks exactly like the phi_SSRGA from Hogan et al. (2017) which is for the backscattering direction only. Hogan et al. (2017) do not seem to ever write their phi_SSRGA as phi_SSRGA(x sin[\theta/2]), but if doing so is correct it would have been clearer if they had done so. If this is correct, you should do so here to make things perfectly clear. And if this is correct, one could then write something as follows:

For the RGA approximation, phi(x*sin[theta/2]) in Eq. 3 is equivalent to

phi_RGA(x,theta) = Your Equation 4.

...

And in the SSRGA approach of Hogan et al. (2027) phi(x*sin[theta/2]) becomes

phi_SSRGA(x*sin[theta/2]) = Your Equation 6

...

Is this correct? If so, then when it comes to Eqs. 8, 9, and 10 later on, the reader will know exactly that it is phi_SSRGA(x*sin[theta/2]) that is to be used in them. If not, perhaps you can see from what is written here the confusion that needs to be cleaned up.

8) Figure 1 looks like it has 35 Dmax bins. If this is correct, then does the database upon which the figure is based have 35 Dmax values for its aggregates? And does each Dmax value lead to an independent set of SSRGA coefficients (or parameters, depending upon what you end up calling them)? If this is correct, then the database upon which Figure 1 is based has 35*kappa, 35*beta, 35*gamma, 35*eta, and 35*alpha_e coefficients. Is this correct? Not sure, especially for alpha_e. Hogan et al. (2017) define alpha_eff on the top right column of their Page 839 as an average over all D / D_max for 50 random orientations of each on n particles of size Dmax and with the radiation moving in the x, y, and z directions, or an average over 50 * 3 * n samples.

If this is the case, then does x in Eq. (7) have a single value for the single averaged value of alpha_e? Generally, one might think of x as a continuously changing variable, but here this may not be the case. Perhaps it is fixed for each Dmax? Anyways, it being fixed would explain why it is not written as part of the functionality in Eqs. 3, 8, 9, and 10. But then to have it as part of the phis is confusing. Perhaps the phis in Eqs. 3 and 6 should be functions of the fixed alpha_e, rather than a presumed continuously changing x? Not sure what is best here, but this is confusing. Clearly written, with a clear recipe as to how to get the coefficients, how to use them to interpolate/extrapolate to new Dmax values, exact units provided for everything, etc..., would be a gift to the community and make this paper an invaluable partner to your algorithms and data bases.

All in all, the paper needs a very clear recipe as to how the SSRGA coefficients (parameters) are generated, how they are used, etc..., perhaps building appropriately on what Hogan et al. (2017) provide in their Section 3.2. With such a clear recipe, this paper gains value, at least to this reviewer.

9) Along the lines of the last part of the previous comment, perhaps Section 3 could also carry along an explicit example of one of the aggregate databases and how it is used to develop a SSRGA approach and then to use the results of the SSRGA approach in an application to arbitrary particle size distributions. Perhaps this could be done with one of the 4 sets of results used in Figure 5. For example, consider LS15 B05. How many Dmax values does this database contain and how many aggregates per Dmax value? Is all of this information in the snowTable? How many SSRGA coefficients are there associated with it? Are these coefficients also stored in the snowTable or are they generated by the SSRGA core module? If by the core module, are they never saved, as they are not listed as part of the Scattering ouput? The results for it in Figure 5 are shown as a continuous curve. But are its results really for discrete values of alpah_eff and hence x? If so, where are these discrete values along the x axis? How is r_eff in Figure 5 related to Dmax or alpha_e? Do all of the

aggregates with the same Dmax in the database have the same mass, so that r_eff is computed directly from this mass? In Figure 2, exactly how are the data products output by SSRGA mapped to the PSD in the radar simulator? Are SSRGA outputs for a particular Dmax mapped/interpolated to corresponding diameter values in the PSD? Making all of this explicit and clear will save interested readers from having to do a lot of work to fill in the details. And making this all clear here in the paper will make using the algorithms perfectly much easier as well.

Please also note the supplement to this comment:
https://gmd.copernicus.org/preprints/gmd-2020-359/gmd-2020-359-RC2-supplement.pdf
* * *
[Figure]

**Supplement:**

[revised manuscript text omitted]

---

## Author Comment (AC1) · 1 Feb 2021

**Reviewer** This paper describes a new modeling system for computing the radiative properties of snow particles in the microwave band. As such, it has the potential to move the field of microwave radiative transfer from its earlier (though still relatively recent) numerical experiments and databases for a limited set of frequencies and particle shapes to a practical community resource that appears to be both easy to use and of wide potential applicability. It includes an impressive database of modeled snow aggregates, both rimed and unrimed, and it utilizes a computationally efficient and flexible

numerical methodology – the self-similar Rayleigh-Gans Approximation (SSRGA) – for the single-particle scattering calculations. I don't know of any other research group undertaking something comparably ambitious and versatile, and I predict that this will quickly become a go-to tool for radiative transfer calculations and as a foundation for inverse methods related to both passive and active microwave remote sensing.

Except for a few specific instances noted below, the paper is well-written and quite thorough in describing both the methods and the limitations of the tool. I visited the github repository and found that the software is convenient to download and install, though I haven't tried using it yet. There's a good start on documentation, though some sections appear not to have been written yet. My overall recommendation is to publish after considering the comments below.

**Author** *Thanks for the very encouraging words and the insightful comments. We agree that the initial code documentation was not exhaustive. We have expanded and completed it in our revised submission.*

**Reviewer** Minor comments:

**Reviewer** lines 54, 131: The SSRGA is introduced here with appropriate citations, but for readers who haven't read those other papers yet, an additional sentence or two explaining what "self-similar" means in this context could be helpful.

**Author** *We have introduced the concept of snowflake self-similarity.*

**Reviewer** line 61: Offhand, at least, I don't know what a "Rayleigh distribution of polarimetric components" is, so maybe a slight elaboration would be useful here as well.

**Author** *The term has been avoided entirely and substituted with a more clear statement about the polarimetric components of RGA Rayleigh scattering.*

**Reviewer** line 77: "parametrized" should be "parameterized"

**Author** *Corrected*

**Reviewer** Eq. (3): The RGA yields a symmetric scattering phase function, as shown by this equation. But I believe that for diameters D (where D is the dimension in the direction of the propagating wave) much greater than about $0.1\lambda$, the phase function quickly shifts toward stronger forward scattering owing to consistently constructive interference in the forward direction (irrespective of size) and varying degrees of destructive interference in the backward direction. Since this is mainly a geometric effect, I'm not even sure whether small $|n - 1|$ eliminates that asymmetry, so I'm wondering whether Eq. (2) tells the whole story. In other words, a particle with kD âĹij 1 or greater, should not, I don't think, conform to Eq. (3) regardless of whether it satisfies Eqs. (1) and (2). If I'm mistaken on this, please disregard this comment, but it would be worth checking and clarifying, if needed.

**Author** *Eq. (3) yields a symmetric scattering phase function only if one does not consider the angular dependency of the form-factor. The sin(theta/2) in the argument of the form-factor in equation (3) takes into account the angular dependency of the phase delays of the various scattered waves from 0 in the forward direction to the maximum (2\*k\*D) in the backward. Of course, if D«$\lambda$, this effect is not relevant and the phase function appears symmetric.*
*The condition on |n-1| does not change this feature and the reviewer is totally correct on this. The point of misunderstanding was on the symmetric nature of the phase function which is actually not symmetric. As pointed out also by Reviewer 2, this passage in the theoretical formulation was unclear and needed to be written more carefully. We hope that the new version helps avoiding possible points of confusion. In particular, we stressed more on the relevance of the angular term in the argument of the form-factor.*

**Reviewer** line 119: Is V the spherical-equivalent volume?

**Author** *We would prefer to avoid the term spherical-equivalent volume since V is the volume of the particle occupied by the dielectric material regardless of its shape. In the case of snowflakes, it is the mass of the snowflake divided by the ice density. We have specified that to avoid confusion.*

**Reviewer** line 148: Fig. 1 is not completely convincing as regards the purported convergence of $\beta$ and $\gamma$. Any curve starts to look flat as it approaches zero on a linear axis. The point might be made more convincingly if a log vertical axis were used in the plot.

**Author** *We have switched to a log vertical axis as suggested.*

**Reviewer** line 467: For what it's worth, Petty and Huang (2010) found that neither Bruggeman nor Maxwell-Garnett dielectric mixing formulas gave the best fit to DDA calculations for soft spheres but rather Sihvola (1989) with an exponent of 0.85.

**Author** *We have computed again the T-matrix solution using the Sihvola generalized mixing formula with the nu parameter set to 0.85 as suggested. The plots and the data have been updated accordingly.*

**Reviewer** General: Several references are made to the computational cost of the DDA method. While true, note that Petty and Huang (2010) demonstrated a variation on the method that at least avoids the extremely large memory requirement of DDSCAT in the case of low density aggregates and effectively allows smaller dense linear systems to be solved rather than very large sparse ones. In other words, I think DDSCAT might not be the ideal benchmark for evaluating the viability of the DDA approach in a resource-limited computational environment. DDA calculations can be run inexpensively on desktop workstations using the alternative approach.

**Reviewer** *We are aware of the Coupled Dipole Approximation (CDA) approach used in Petty and Huang (2010). The DDA implementation we used is actually ADDA and not DDSCAT. ADDA implements a "SPARSE" mode which, despite its name, is analogous to the CDA method https://github.com/adda-team/adda/issues/98. Davide Ori also explored the possibility to leverage on the low memory footprint of the method to accelerate the computations further using GPU computing http://amsdottorato.unibo.it/7521/ https://github.com/adda-team/adda/tree/sparse_ocl*
*Although the approach is very interesting for the mentioned application (computing*

*the microwave scattering properties of low-density aggregates) it becomes not feasible when the number of scattering elements is large. This is a problem for our application that involves either rimed particles (high density) or high frequency (increased require-ments with respect to particle shape resolution).*

*Despite the memory footprint being low, the computational complexity of CDA is O(n\*\*2), where n is the number of volume elements composing the aggregate shape. This leads to a rapid increase in the computing time needed to solve the system of linear equations as the number of scattering elements increases.*

---

## Author Comment (AC2) · 1 Feb 2021

**Reviewer** The paper's primary purpose is to present and describe snowScatt 1.0, a toolkit for computing the scattering properties of unrimed and rimed aggregates from microwave to sub-millimeter electromagnetic wavelengths using the Self-Similar Rayleigh Gans Approximation (SSRGA). A secondary purpose of the paper is to illustrate the value of the SSRGA in the first place. Review:
The paper is certainly relevant to GMD, and the toolkit will be of value to the community in the years to come. Given the primary purpose of the paper, Sections 2 and 3 are

most important and must be written perfectly, providing all of the details necessary to understand the code in detail and to apply it without error. In this regard the paper needs more work, and the focus of this review is on these two sections. Because there are no major issues with the approach, the required revisions suggested for the paper are listed as minor in nature, though certainly required in order to maximize impact of the paper. McCusker, Westbrook, and Tynella have just come out with a paper in QJRMS with results that address some of the issues raised in this paper. The reference for this paper is An accurate and computationally cheap microwave scattering method for ice aggregates: the Independent Monomer Approximation Karina McCusker, Chris Westbrook, and Jani Tyynelä By incorporating relevant results from McCusker et al., e.g., problems with forward scattering in the RGA, the authors will strengthen this paper as well. Finally, all qualitative comparative language should be removed from the paper and replaced with quantitative measures. This will not be hard to do because quantitative measures appear to be readily available where qualitative language is used. The PDF file returned to the authors contains mark-up using Acrobat Comments that indicates some of these places. The mark-up also contains other issues the authors might consider addressing. Detailed Comments on Sections 2 and 3: 0) For an algorithm/data product paper such as this one, every parameter in every algorithm and data product must have clearly specified units. Lacking clear, correct, and internally consistent units, things will get confusing in a hurry.

**Author** *Thanks for the comments. We have followed your suggestion for improving the clearness and general quality of the paper with a special attention to sections 2 and 3. Both the review and the detailed annotated manuscript have been taken into account for the revision. The paper from McCusker et al. is certainly relevant and it is very fortunate that it has been published before the conclusion of this review process. We replaced all qualitative discussion on the method accuracy with quantitative statements and defined each quantity with its measuring units. Thanks also for the review of the grammar and spelling you have provided.*

**Reviewer** 1) Calling sigma on Line 118 an intensity is confusing. Intensity in the SI system has units of W / sr. Sigma on this line seems to have units of $m^2 sr^{-1}$ based on Eq. (8). (Eq. 8 is an integral over all 4*pi steradians of solid angle and the result is in $m^2$.) So, it is probably better called a differential scattering cross section if it does indeed have units of $m^2 sr^{-1}$.

**Author** *True, this is certainly a differential scattering cross section, we have avoided referring to it as "intensity".*

**Reviewer** 2) Why have the word "Rayleigh" in front of "dielectric factor" on Line 119?

**Author** *This is an error, we have corrected the text.*

**Reviewer** 3) The form factor, itself, has nothing to do with deviations from any quantity. As one factor in a bunch of factors, it might be interpreted as creating deviations in the product of other factors, but to present it as such is to hide its own direct physical significance. So, much of the sentence on Line 120 is not helpful in the context of this paper. Lines 121 and 122, together with Eq. 4, make this clear. For example, there is no deviation involved in Eq. 4, just phase changes as a result of location and direction. Simple and clear these lines are.

**Author** *We agree that the form factor should be presented as integration of phase delays and we changed our discussion accordingly.*

**Reviewer** 4) Is phi in Eq. 3 the same thing as phi_RGA in Eq. 4? If so, please give them the same label. If not, explain the difference between the two. If the same thing, phi_RGA should be a function of x and theta, similar to phi in Eq. 3, and as stated on Line 125.

**Author** *Yes, the reviewer is right on this point. We made the notation consistent. See also point 7)*

**Reviewer** 5) Line 128, Eq. (5), and Line 129 are out of context. Probably best to move them to just before Eq. 8.

[Figure]

**Author** *We moved the paragraph as suggested*

**Reviewer** 6) On Line 12 of the Abstract reference is made to "the set of SSRGA coefficients", and these words occur elsewhere throughout the text. What these coefficients represent is never made perfectly clear. Are these the "five parameters (Hogan et al., 2017, alpha_e, kappa, beta, gamma, zeta_1)" that occur on Lines 132 and 133? If so, make this perfectly clear and always use the same words (e.g., "coefficients" or "parameters") to describe them. Otherwise, things get confusing in regards to them.

**Author** *We opted for the word "parameters" as they do not appear in the formulation as multiplicative coefficients. We also made a clear statement about the fact that we always refer to the collection of them generically as the "set of SSRGA parameters".*

**Reviewer** 7) The shape and orientation of any particle to which Eq. 4 is applied are both captured by the integral over V, i.e., where the ice volume elements are actually located. Line 125 correctly characterizes this state-of-affairs. From this perspective, Lines 131 and 132 need more clarity because as currently written Eq. 6 has no dependence on theta. Eq. 6 looks exactly like the phi_SSRGA from Hogan et al. (2017) which is for the backscattering direction only. Hogan et al. (2017) do not seem to ever write their phi_SSRGA as phi_SSRGA(x sin$(\theta/2)$),

but if doing so is correct it would have been clearer if they had done so. If this is correct, you should do so here to make things perfectly clear. And if this is correct, one could then write something as follows:
For the RGA approximation, $phi(x * sin(\theta/2))$ in Eq. 3 is equivalent to $phi\_RGA(x, \theta)$ = Your Equation 4. ... And in the SSRGA approach of Hogan et al. (2027) $phi(x * sin(\theta/2))$ becomes $phi\_SSRGA(x * sin(\theta/2))$ = Your Equation 6 ...
Is this correct? If so, then when it comes to Eqs. 8, 9, and 10 later on, the reader will know exactly that it is $phi\_SSRGA(x * sin(\theta/2))$ that is to be used in them.
If not, perhaps you can see from what is written here the confusion that needs to be cleaned up.

**Author** *Thanks for pointing this out. Indeed the change in the argument number required some clarification. We tried to make things more clear by unravelling the math behind equations 3 and 6. First, we corrected equation 3 by removing the dependence to sin(theta/2); this is because (as pointed by Hogan 2017) the scaling of the path delays as sin(theta/2) is not applicable in general. We then introduced the scaled size parameter x_theta which explicitly takes into account this geometrical simplification. We substituted then the arguments (x, theta) of phi_ssrga with the unique argument x_theta. We hope this will make the mathematics more clear. It also helped the discussion about the phase function being asymmetric towards the forward scattering direction as this was a point of misunderstanding with reviewer 1*

**Reviewer** 8) Figure 1 looks like it has 35 Dmax bins. If this is correct, then does the database upon which the figure is based have 35 Dmax values for its aggregates? And does each Dmax value lead to an independent set of SSRGA coefficients (or parameters, depending upon what you end up calling them)? If this is correct, then the database upon which Figure 1 is based has 35*kappa, 35*beta, 35*gamma, 35*eta, and 35*alpha_e coefficients. Is this correct? Not sure, especially for alpha_e. Hogan et al. (2017) define alpha_eff on the top right column of their Page 839 as an average over all D / Dmax for 50 random orientations of each on n particles of size Dmax and with the radiation moving in the x, y, and z directions, or an average over 50 * 3 * n samples.

**Author** *The reviewer interpretation is correct. We derive SSRGA parameters (including effective aspect ratio) for different size bins (35 for the aggregates taken into account in Figure 1). We define the orientations of the particles as they are produced by the aggregation model. The SSRGA parameters included in the database are derived for vertical incident orientation, (we have made this point more clear in text). For a zenith incident direction the azimuth direction does not matter. Using the snowScatt tool, it is possible to derive SSRGA parameters also for different incident angles. When an off-zenith incident direction is taken into account also azimuth orientation becomes*

[Figure]

*relevant and the code allows to sample the shapefile for multiple azimuth angles. It is also possible to derive the SSRGA parameters for random incident angles, similarly to what Hogan (2017) did. The example folder of snowScatt contains an example of how to derive such parameters.*

**Reviewer** If this is the case, then does x in Eq. (7) have a single value for the single averaged value of alpha_e? Generally, one might think of x as a continuously changing variable, but here this may not be the case. Perhaps it is fixed for each Dmax? Anyways, it being fixed would explain why it is not written as part of the functionality in Eqs. 3, 8, 9, and 10. But then to have it as part of the phis is confusing. Perhaps the phis in Eqs. 3 and 6 should be functions of the fixed alpha_e, rather than a presumed continuously changing x? Not sure what is best here, but this is confusing. Clearly written, with a clear recipe as to how to get the coefficients, how to use them to interpolate/extrapolate to new Dmax values, exact units provided for everything, etc..., would be a gift to the community and make this paper an invaluable partner to your algorithms and data bases. All in all, the paper needs a very clear recipe as to how the SSRGA coefficients (parameters) are generated, how they are used, etc..., perhaps building appropriately on what Hogan et al. (2017) provide in their Section 3.2. With such a clear recipe, this paper gains value, at least to this reviewer.

**Author** *We agree with the reviewer on the fact that this paper needs a clear and linear description of how the various parameters are derived and applied for the calculation of the scattering properties. We think that our revised section 2.2 will help understanding the algorithms implemented in snowScatt. In particular, the contribution of alpha_eff is made clear by what is now eq. (7).*

**Reviewer** 9) Along the lines of the last part of the previous comment, perhaps Section 3 could also carry along an explicit example of one of the aggregate databases and how it is used to develop a SSRGA approach and then to use the results of the SSRGA approach in an application to arbitrary particle size distributions. Perhaps this could be done with one of the 4 sets of results used in Figure 5. For example, consider LS15 B05. How many Dmax values does this database contain and how many aggregates per Dmax value? Is all of this information in the snowTable? How many SSRGA coefficients are there associated with it? Are these coefficients also stored in the snowTable or are they generated by the SSRGA core module? If by the core module, are they never saved, as they are not listed as part of the Scattering output? The results for it in Figure 5 are shown as a continuous curve. But are its results really for discrete values of alpha_eff and hence x? If so, where are these discrete values along the x axis? How is r_eff in Figure 5 related to Dmax or alpha_e? Do all of the aggregates with the same Dmax in the database have the same mass, so that r_eff is computed directly from this mass? In Figure 2, exactly how are the data products output by SSRGA mapped to the PSD in the radar simulator? Are SSRGA outputs for a particular Dmax mapped/interpolated to corresponding diameter values in the PSD? Making all of this explicit and clear will save interested readers from having to do a lot of work to fill in the details. And making this all clear here in the paper will make using the algorithms perfectly much easier as well.

**Author** *Each snowTable carries the size-resolved SSRGA parameter along with power-law fitting parameters for the microphysical properties (mass and area). We have provided two different examples of how to use the snowScatt library in Figs 5 and 6. To improve the readers comprehension we expanded the description of the snowScatt setup used to generate those figures. We think that these two examples, together with the expanded description are sufficient to explain in detail how to use snowScatt and the code internal workflow. Nonetheless, as stated in the Code and Data Availability section both the development version of the code and the packaged zenodo version carry along the scripts that have been used to generate any figure included in the manuscript.*

*The number of Dmax values in the snowTables depends on the particle type. In general we used tables with Dmax bins evenly spaced by 1 mm. Given the large number of shapes included in the snow types of the CaE database the snowTable*
*resolution is increased to 0.5 mm. The actual number of size bins depend on the range of sizes spanned by the aggregate type. The number of aggregates per bin depends on the total number of aggregates and the number of bins. The minimum number of particle shapes per bin is 15 which occur for the tables corresponding rimed aggregates with ELWP=2.0. We have expanded the description of the snowTable in section 3.1 to make this point more clear.*

*The snowTables carry along the size resolved SSRGA parameters and some additional information stored as metadata. We have already seen the opportunity to add more strict rules on metadata, including the number of aggregates per size bin (see, for example the issue open on the development page of snowScatt already in December 2020 https://github.com/OPTIMICe-team/snowScatt/issues/9).*
*This feature will certainly become part of the code in the future, but it was not considered a priority since these additional informations are not strictly required for the computation of scattering properties using SSRGA.*

*The only independent variable is Dmax, the SSRGA parameters (kappa, gamma, beta, zeta and alpha_eff) are derived from the size-resolved table of coefficients using nearest-neighbour interpolation. Mass is generally computed using the power-law coefficients stored in the snowTable (as it is done in Fig 5), but it can also be modified (as in Fig 6) in order to match a particular value, in this case, the code still uses the SSRGA parameters fitted from the snowTable. We have added more explanations about this procedure in the text.*

*Since the alpha_eff values are obtained through nearest-neighbour interpolation one can say that they are discrete, however, D_max and mass are continuously changing along the curve and the discretization done in the snowTable is sufficiently fine to not show any "jump" in the curves.*

[Figure]

*As already said the only "true" independent variable is Dmax, which means that it is only necessary to define a common set of maximum sizes to map the output of each module onto the same domain. We made this point more clear by stating explicitly what are the input parameters of each module in figure 2 and in the text.*